# SAFA-SNN: Sparsity-Aware On-Device Few-Shot Class-Incremental Learning with Fast-Adaptive Structure of Spiking Neural Network

**Huijing Zhang**[1]     **Muyang Cao**[1,2]     **Linshan Jiang**[3*]     **Xin Du**[1*]     **Di Yu**[1]
**Changze Lv**[4]     **Shuiguang Deng**[1]
[1] Zhejiang University     [2] Shanghai Innovation Institute
[3] Southern University of Science and Technology     [4] Fudan University
{huijingzhang, muyangc, xindu, yudi2023, dengsg}@zju.edu.cn
linshan@nus.edu.sg   czlv24@m.fudan.edu.cn

## ABSTRACT

Continuous learning of novel classes is crucial for edge devices to preserve data privacy and maintain reliable performance in dynamic environments. However, the scenario becomes particularly challenging when data samples are insufficient, requiring on-device few-shot class-incremental learning (FSCIL). Although existing work has explored parameter-efficient FSCIL frameworks based on artificial neural networks (ANNs), their deployment is still fundamentally constrained by limited device resources. Spiking neural networks (SNNs) process spatiotemporal information efficiently, offering lower energy consumption, greater biological plausibility, and compatibility with neuromorphic hardware than ANNs. In this work, we propose an SNN-based method containing Sparsity-Aware neuronal dynamics and Fast Adaptive structure (SAFA-SNN) for on-device FSCIL. By threshold regulation, most neurons exhibit stable spikes and others exhibit adaptive spikes. As a result, synaptic traces that encode base-class knowledge are naturally preserved, thereby alleviating catastrophic forgetting. To cope with spike non-differentiability in backpropagation, we employ a gradient-free technique, i.e., zeroth-order optimization. Moreover, class prototypes can limit overfitting on few-shot data but introduce bias. We enhance prototype discriminability by orthogonal subspace projection. Extensive experiments conducted on two standard benchmark datasets (CIFAR-100 and Mini-ImageNet) and three neuromorphic datasets (CIFAR10-DVS, DVS128 Gesture, and N-Caltech101) demonstrate that SAFA-SNN outperforms baselines, specifically achieving at least 4.01% improvement at the last incremental session on Mini-ImageNet and 20% lower energy cost on CIFAR-100 over baselines with practical implementation. [1]

## 1 INTRODUCTION

Intelligent edge devices sense dynamic contexts in their surrounding environment and make decisions based on sensor data (Ravaglia et al., 2021). The high labeling cost and privacy constraints make it difficult to acquire sufficient data for continual batch learning (Wang et al., 2022b). To this end, on-device Few-Shot Class-Incremental Learning (FSCIL) is introduced to incrementally learn from newly collected sensory data with scarce labeled samples while preserving existing knowledge. Spiking Neural Networks (SNNs) (Maass, 1997) are attractive for edge devices (Deng et al., 2025), as neurons fire only upon event-driven activation, reducing unnecessary computation cost (Deng et al., 2020; Lv et al., 2024).

---

[*]Corresponding authors.
[1]Codes are available at https://github.com/ZhangHuiJing2020/SAFA-SNN.

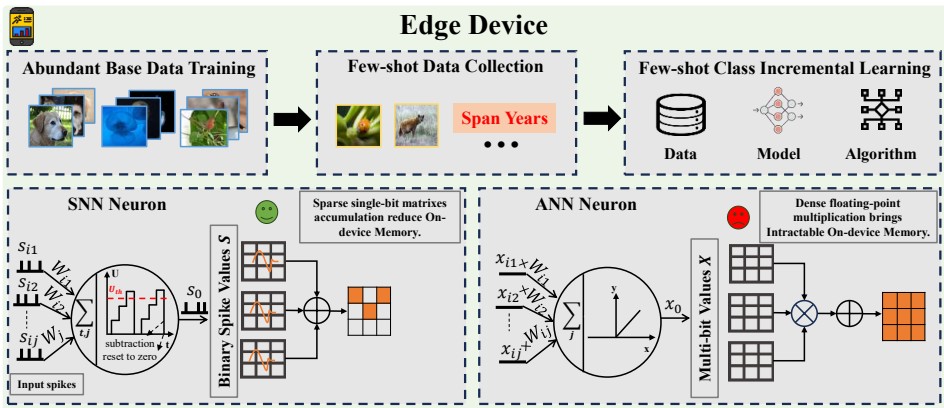

Figure 1: ***Upper.*** The on-Device FSCIL scenario includes three stages: Abundant base data training, Few-shot data collection, and Few-shot class-incremental learning. ***Bottom.*** Compared to the ANN neuron, the SNN neuron communicates by spikes, showing memory and computation benefits.

However, on-device FSCIL faces two significant challenges, i.e., catastrophic forgetting (French, 1999) and overfitting (Tao et al., 2020). Catastrophic forgetting reflects the model's difficulty learning new tasks while retaining information of originally learned tasks, necessitating a delicate balance between plasticity and stability. Overfitting occurs when the model memorizes limited training samples, leading to poor generalization. On the one hand, recent advanced solutions tackle these challenges by parameter-efficient fine-tuning (PEFT) (Liu et al., 2024a), which fine-tunes parameters on top of a large frozen pre-trained model. Nonetheless, these approaches remain ineffective under limited memory budgets (i.e., 4–12 GB) (Ma et al., 2023). On the other hand, concurrent on-device SNNs concentrate on integrating SNN algorithms with specialized neuromorphic hardware, which shifts the consideration of the above problems to the later model. Additionally, most existing neuromorphic systems are still trained offline and remain static during development (Safa, 2024), presenting a heavy dependency on abundant labeled data.

Besides, smart devices are severely constrained in terms of memory capacity and maximum performance (Shuvo et al., 2022), extremely scarce labeled samples, and the persistent need for model updates. Edge devices are incapable of accumulating comprehensive data on all classes due to limited storage capacity and must simultaneously perform low-power inference while continuously learning from real-time data streams. We provide the process of on-device FSCIL scenario and compare the behavior of SNN and ANN neuron in this scenario in Figure 1.

To tackle the above challenges, we propose Sparsity-Aware neuronal dynamics and Fast-Adaptive prototype subspace projection with SNN (SAFA-SNN) for general on-device FSCIL. Sparsity measures the spike frequency of neurons. Neurons with dynamic thresholds are defined as stable or active, where most remain stable and only a few are active for new tasks, minimizing updates to the majority of synapses. Zeroth-order optimization is a gradient-free method to approximate true gradients using only function values (Liu et al., 2020b), which enables SNNs to handle non-differentiable spike activation while avoiding limited width of surrogate gradients (Lian et al., 2023). In incremental learning sessions, retraining the model with few-shot samples is inherently prone to severe overfitting. To migrate this issue, we project the prototypes onto the subspace spanned by base classes, enhancing the discriminability of new classes. We conduct practical implementation and extensive experiments on static and neuromorphic datasets, demonstrating the performance of SAFA-SNN.

The main contributions of this paper are as follows:

- We emphasize practical challenges of on-device FSCIL, where both training and inference are performed on resource-constrained devices using energy-efficient, hardware-friendly SNNs, contrasting with ANN-based FSCIL that depends on large-scale server models.

- We propose a sparsity-aware FSCIL method, called SAFA-SNN, with three key features. First, incorporating spikes for prediction, we propose the Sparsity-Aware dynamics for SNN. Then, to solve the problem of non-differentiable spikes in back propagation, we adopt zeroth-order optimization within neuronal dynamics. Finally, we enhance the dis-

criminability of prototypes by subspace projection in incremental learning. To the best of our knowledge, this is the first SNN-based solution towards general on-device FSCIL.

- To prove the effectiveness of our proposed SAFA-SNN for on-device FSCIL, we implement SAFA-SNN on realistic devices (i.e., Jetson Orin AGX), and evaluate on five datasets (CIFAR-100, MiniImageNet, CIFAR10-DVS, DVS128 Gesture, and N-Caltech101), and various models (Spiking VGG, Spiking ResNet, and Spikingformer), demonstrating that SAFA-SNN achieves excellent performance and notable sparsity advantages. We further evaluate the actual on-device energy consumption, demonstrating our efficiency advantage.

## 2 RELATED WORK

### 2.1 FEW-SHOT CLASS-INCREMENTAL LEARNING

To the best of our knowledge, existing methods for few-shot class-incremental learning (FSCIL) are mostly ANN-based, with little exploration in SNNs. The primary work (Tao et al., 2020) preserves feature topology using a neural gas network. Researchers (Zhang et al., 2021) model prototypes as nodes in graph attention network to propagate context information. Existing methods (Zhou et al., 2022; Wang et al., 2023) commonly adopt prototypes to replace traditional MLPs and refine them (Song et al., 2023). Some studies (Shi et al., 2021; Liu et al., 2023) tackle class imbalance by exploring optimal base class distributions. Recent advances (Park et al., 2024; Sun et al., 2024; Wang et al., 2024) train a few parameters (prompts) from the frozen pre-trained model, achieving high accuracy. However, memory overhead limits their use for resource-constrained devices. Prior work focuses on a specific processor (Huo et al., 2025), while we explore FSCIL on general devices.

### 2.2 SPIKING NEURAL NETWORKS WITH DYNAMIC THRESHOLD

In the realm of SNNs, the dynamic threshold mechanism aims to manipulate the threshold spontaneously. BDETT computes thresholds via average membrane potentials for neuronal homeostasis (Ding et al., 2022), but excessive spiking remains. Highly active neurons skew the average, causing aggressive firing and reduced sparsity. LTMD learns initial membrane potentials for heuristic neuron thresholding (Wang et al., 2022a), but neuron behaviors across layers remain interdependent in spatial and temporal processing. Soft threshold schemes enable adaptive layer-wise sparsity (Chen et al., 2023), but independent pruning causes error accumulation, affecting overall model performance. Efforts with combinations of adaptive threshold methods have led to innovative approaches (Wei et al., 2023; Hao et al., 2024). Our paper further tackles FSCIL challenges via sparsity-aware dynamic thresholds that shift networks to task-specific local contexts.

### 2.3 ON-DEVICE SNN TRAINING AND INFERENCE

Previous studies on on-device SNNs can be classified into two categories: inference and training with updating. The lightweight SNN is a crucial application for resource-constrained edge devices (Yu et al., 2024a). For instance, Lite-SNN (Liu et al., 2024b) proposes a spatial-temporal compression strategy to reduce memory and computational cost. Most researchers focus on efficient quantization for SNN-training like SNN pruning (Wei et al., 2025; Qiu et al., 2025; Yu et al., 2025), to facilitate efficient deployment. Current SNN training and updating focus on gradient update (Anumasa et al., 2024) or local SNN training optimization (Mukhoty et al., 2023; Yu et al., 2024b) to enable low-latency SNN training on devices. In this work, we explore both SNN-batch training on abundant base classes and inference in FSCIL phases, which typically occur in the overchanging on-device environments, achieving high performance and low energy consumption.

## 3 PRELIMINARIES

### 3.1 SPIKING NEURAL NETWORK

We use the leaky integrate-and-fire (LIF) neuron model, a simplified but common approach to neuromorphic computing that mimics biological neurons through charging, leakage, and firing. The

process of state updating in LIF neurons follows certain principles:

$$\mathbf{U}'_t = \gamma \mathbf{I}_t + \mathbf{U}_{t-1}, \mathbf{I}_t = f(\mathbf{x}; \boldsymbol{\theta}), \tag{1}$$

$$\mathbf{S}_t = H(\mathbf{U}'_t - \mathbf{U}_{\text{th}}), \tag{2}$$

$$\mathbf{U}_t = \mathbf{S}_t \mathbf{U}_{\text{reset}} + (\mathbf{1} - \mathbf{S}_t)\mathbf{U}'_t, \tag{3}$$

where $H = \mathbf{1}_{\mathbf{U}_{t'} > \mathbf{U}_{\text{th}}}$, $\gamma$ is the time constant, $\mathbf{I}_t$ is the spatial input calculated by applying function $f$ with $\mathbf{x}$ as input and $\theta$ as learnable parameters, and $\mathbf{U}'_t$ is a temporary voltage at time step $t$, whose value instantly changes. At time step $t$, when the membrane $\mathbf{U}'_t$ rises and reaches the threshold $\mathbf{U}_{\text{th}}$, a spike denoted by $\mathbf{S}_t$ is generated as 1, and $\mathbf{U}'_t$ will be reset to $\mathbf{U}_{\text{reset}}$; otherwise, $\mathbf{S}_t$ remains 0. Afterwards, $\mathbf{U}'_t$ changes to $\mathbf{U}_t$. SNNs present challenges in terms of the non-differentiable activation function in Equation (2). The backpropagation of directly training SNNs can be described as

$$\frac{\partial L}{\partial \mathbf{W}} = \sum_{t=1}^{T} \frac{\partial L}{\partial \mathbf{S}_t} \frac{\partial \mathbf{S}_t}{\partial \mathbf{U}_t} \frac{\partial \mathbf{U}_t}{\partial \mathbf{I}_t} \frac{\partial \mathbf{I}_t}{\partial \mathbf{W}}, \tag{4}$$

where $\mathbf{W}$ is the weight of input, and $\frac{\partial \mathbf{S}_t}{\partial \mathbf{U}_t}$ is non-differentiable, which is typically replaced by surrogate gradient (SG) methods (Wu et al., 2018). However, SG deviates significantly from the true gradients and easily causes vanishing gradient problem (Neftci et al., 2019). To address this issue, we employ zeroth-order optimization to estimate the gradients, inspired by Mukhoty et al. (2023).

## 3.2 CLASS PROTOTYPE

Class prototypes are widely used in few-shot learning based on the observation by Snell et al. (2017). In FSCIL, researchers use prototypes as dynamic classifiers (Zhang et al., 2021). In detail, they decouple the FSCIL model into a feature encoder $\phi_\theta(\cdot)$ with parameters $\theta$ and a linear classifier $W$, and freeze $\phi_\theta(\cdot)$ during incremental sessions, with only the classifier being updated. For each new class $i$, the weight $W_i$ in the classifier is parameterized by the mean embeddings of each class, i.e., the prototype $P_i$, which can be denoted as $P_i = \frac{1}{K} \sum_{j=1}^{|D_s|} \mathbb{I}_{\{y_j = i\}} \phi(x_j)$, where $K$ denotes the number of samples in the class $i$.

## 4 ON-DEVICE FSCIL PROBLEM

On-device FSCIL problem depicts data with scarce samples always coming, models need to maintain performance on all seen classes while maintaining low latency and high security, enhanced data privacy, and real-time decisions under resource-limited edge devices. The data resources are always scarce, for example, users take an average of 4.9 photos per day (Gong et al., 2024).

Assuming FSCIL tasks contain a base session and a sequence of $S$ incremental sessions, the training data available for each session can be denoted as $\{D_0^{train}, D_1^{train}, ..., D_S^{train}\}$. $D_s^{train} = \{(x_i^s, y_i^s)\}_{i=0}^{|D_s^{train}|}$, where $x_i^s \in X_s$ is the training instance and $y_i^s \in Y_s$ is its corresponding label in session $s$. The classes in each session $s$ is disjoint with another session $s'$, i.e., $D_s^{train} \cap D_{s'}^{train} = \emptyset$. The training data in session 0 have abundant instances while each incremental session has much smaller instances, i.e., $|D_0^{train}| >> |D_{s>0}^{train}|$. Each incremental session has training sets in the form of N-way K-shot, i.e, there are N classes and each class has K samples. In session $s$, the model's performance is assessed on validation sets from all encountered datasets ($\leq s$) to minimize the expected risk $\mathcal{R}(f, s)$ over all seen classes (Zhang et al., 2025):

$$\mathbb{E}_{(x_i, y_i) \sim \mathcal{D}_0^{train} \cup \cdots \cup \mathcal{D}_s^{train}} \left[ \mathcal{L} \left( f(x_i; \mathcal{D}_s^{train}, \theta^{s-1}), y_i \right) \right], \tag{5}$$

where the algorithm $f$ constructs a new model using current training data $\mathcal{D}_s^{train}$ and parameters $\theta^{s-1}$ to minimize the loss $\mathcal{L}$. The general goal is to continually minimize the risk of each new session, i.e., $\sum_{s=0}^{S} \mathcal{R}(f, s)$. Note that in this paper, "session" and "task" are used interchangeably.

In many real-world scenarios, when FSCIL applications are deployed on resource-limited edge devices, the learning efficiency w.r.t. both few-shot inference speed and memory footprint becomes a crucial metric. Being able to quickly adapt deep prediction models in low computational cost at the edge is necessary to better suit the needs of personal mobile users. However, these aspects are rarely explored in prior FSCIL research, which motivates us to solve these on-device FSCIL challenges beyond catastrophic forgetting and overfitting.

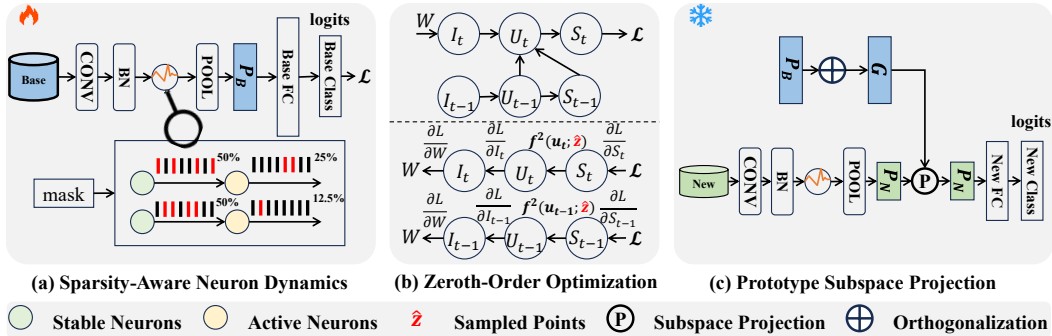

Figure 2: SAFA-SNN framework includes three main components: (a) Training abundant data and selecting active and stable neurons by masks. (b) Top: forward propagation through FSCIL process; Bottom: backpropagation using zeroth-order optimization only in the base class training. (c) Freezing backbones and updating the prototypes by subspace projection in the incremental inference.

## 5 METHODOLOGY

### 5.1 MOTIVATION

While considerable research focuses on soft-subnetworks that freeze the major part of the model parameters to preserve learned knowledge and train a minor part of parameters for the selected subnetwork in incremental learning sessions (Mazumder et al., 2021; Kang et al., 2023), inspired by the regularized lottery ticket hypothesis, the question of expanding the network structure remains unsolved. Additionally, biological neural networks are interconnected through synapses, receiving input spikes and emitting output spikes according to the membrane accumulation, forming different subnetworks by synapse weights, but different regularized subnetworks formed by different spiking firing dynamics in neurons remain unexplored. Motivated by these challenges, we explore if the regulation of neuronal dynamics can help the trade-off between plasticity and stability in dealing with FSCIL tasks. Despite ongoing efforts, overfitting remains a challenge. Subspace projection, using class-specific matrices, offers robustness in few-shot learning (Simon et al., 2020). Despite previous studies explore prototype rectification methods by trainable networks (Tang et al., 2024) and training-free calibration (Wang et al., 2023), the prototypes of novel classes still tend to deviate to the base one. These motivate the need for the subspace projection scheme that represents each new class with an orthogonal subspace basis and projects onto the most similar base classes.

### 5.2 OVERVIEW OF SAFA-SNN

SAFA-SNN for on-device FSCIL comprises three parts: a sparsity-aware dynamic mechanism to mitigate forgetting, the formulation of zeroth-order optimization for non-differentiable spikes, and a prototype subspace projection for fast adaptation on few-shot new class. Figure 2 illustrates the overview of SAFA-SNN in FSCIL, and the full FSCIL procedure is detailed in Algorithm 1.

### 5.3 SPARSITY-AWARE NEURONAL DYNAMICS

Previous studies have demonstrated the functional role of neuronal firing activity in visual classification learning (Freedman et al., 2002). The sparsity-aware neuronal dynamics implement efficient heterogeneous plasticity in SNNs by a dynamic threshold function. It involves neuron division and dynamic threshold mechanism built on the LIF neuron model. Following a homeostatic plasticity rule (Turrigiano, 2012), stable neurons maintain firing rates that stay close to the firing rate during base-class training, which helps preserve previously acquired knowledge. Neurons in each channel are randomly designated as either stable or adaptive, using a channel-wise mask $\mathbf{M}$, which is defined as $\mathbf{M} = \{\mathbf{m}_c\}, c = 1, 2, ..., C$, where $C$ is the channel number and $\mathbf{m}_c = \mathbf{1}_{c \leq \lfloor \eta C \rfloor}$, where $\eta \in (0, 1)$ is the adaptive ratio, and neurons in channel $c$ are adaptive if $\mathbf{m}_c = \mathbf{1}$, otherwise they are stable. Most neurons remain stable to maintain consistent spike patterns, while others adapt to incremental learning. We define the sparsity as channel-wise firing rate with spike output $\boldsymbol{S}$ as

---

**Algorithm 1** Overall SAFA-SNN Algorithm

---

**Input**: Dataset $\{D_s\}_{s=0}^S$, membrane $\mathbf{U}$, feature extractor $\phi$
**Output**: Spike trains $\mathbf{S}_t$, parameters $\boldsymbol{\theta}$
 1: Initialize mask $\mathbf{M}$ to set $\eta\%$ adaptive neurons 1 and setting others 0 in each channel
 2: Initialize threshold $\mathbf{U}_{\text{th}}$ with zeros
 3: **for** session $s = 0$ to $S$ **do**
 4:     Conduct state updating in LIF neuron models by Equation (1), (2), and (3)
 5:     **if** $s == 0$ **then**
 6:         Calculate the threshold regulation factor $\mathbf{A}$ by Equation (6)
 7:         Sample point $z_i \sim \mathcal{N}(0,1), i = 1, 2, ..., b$
 8:         **if** $|x| < \delta|z_i|$ **then**
 9:             Estimate the gradient $\hat{g}_i = m_i \cdot |z_i|$
10:         **end if**
11:         Estimate the non-differential $\frac{\partial S_t}{\partial \mathbf{U}_t}$ by Equation (10)
12:         Conduct base session optimization $L$ and update $\phi(\theta)$ by Loss in Equation (13)
13:     **else**
14:         Update threshold $\mathbf{U}_{\text{th}}$ by $\mathbf{U}'_{\text{th}} = \mathbf{U}_{\text{th}} + \mathbf{A}(\mathbf{r}_c - \mathbf{r}_b)$
15:         Compute the projection subspace by Equation (14)
16:         Conduct incremental learning with $\tilde{\mathbf{P}}$ in Equation (15)
17:     **end if**
18: **end for**

---

$\mathbf{r} = \frac{1}{|\Omega|} \sum_{(b,t,n) \in \Omega} \boldsymbol{S}(b,t,n)$. Define the threshold regulation factor as

$$\mathbf{A} = \beta(\mathbf{1} - \mathbf{M}) + \gamma\mathbf{M}, \tag{6}$$

where $\beta$ and $\gamma$ are hyper-parameters, and setting $\beta > \gamma$ ensures that thresholds of adaptive neurons change more freely than those of stable neurons. The threshold is dynamically updated by

$$\mathbf{U}'_{\text{th}} = \mathbf{U}_{\text{th}} + \mathbf{A}(\mathbf{r}_c - \mathbf{r}_b), \tag{7}$$

where $\mathbf{r}_c$ and $\mathbf{r}_b$ are the sparsity of neurons in current incremental task and base task 0, respectively. With dynamic thresholds, neurons maintain relatively stable spike patterns, which limits unnecessary synaptic updates. As a result, synaptic traces encoding base-class knowledge are preserved. The whole neuronal dynamics process is provided in Appendix A.

## 5.4 ZEROTH-ORDER OPTIMIZATION

**Multi-point estimate.** Zeroth-order optimization is beneficial to computation-difficult or infeasible gradient problems, since it can approximate the full gradients or stochastic gradients through function value (Liu et al., 2020b). We start by presenting the gradient estimate of $f(x)$ with two-point directional derivative as

$$\hat{\nabla}f(x) := \frac{\phi(d)}{\mu} \Sigma_{i=1}^b [f(x + \delta\mathbf{z}_i) - f(x - \delta\mathbf{z}_i)], \tag{8}$$

where $b$ is the number of i.i.d. samples $\{\mathbf{z}_i\}_{i=0}^b$, and $d$ is a dimension-dependent factor. Let $g(u)$ denotes the central finite difference approximation $\frac{\partial \mathbf{S}_t}{\partial \mathbf{U}_t}$ with membrane potential $u = u_t - u_{th}$. Then, for $u \in \mathbb{R}^n$,

$$g^2(u; \delta, z) = \frac{H(u + \delta\boldsymbol{z}) - H(u - \delta\boldsymbol{z})}{2\delta} \boldsymbol{z} = \begin{cases} 0, |u| > \delta|\boldsymbol{z}| \\ \frac{|\boldsymbol{z}|}{2\delta}, |u| < \delta|\boldsymbol{z}| \end{cases}, \tag{9}$$

where $\boldsymbol{z}$ obeys a specific distribution $p$ over $b$ empirical samples $\{z_i\}_{i=0}^b$ and $H$ is the heaviside function defined in Section 3.1. Assessing whether perturbations $u + z\delta$ and $u - z\delta$ induce a spike in the neuron allows for an approximation of the gradient. Through random permutation, $g^2(u)$ accommodates the influence of all neighbors, for a balanced depiction of local dynamics. The approximation to $\hat{\nabla}g(x)$ at a given input $\mathbf{u}$ can be computed as

$$\frac{\partial \mathbf{S}_t}{\partial \mathbf{U}_t} := \frac{1}{b} \sum_{i=1}^b g^2(\mathbf{u}; \delta, \mathbf{z}_i). \tag{10}$$

Gradients are typically approximated by averaging multiple perturbed samples, where each perturbation produces a local gradient, resulting in a more reliable estimate (See Appendix B).

**Convergence analysis of ZOO.** We start with some conventional criteria for convergence analysis in zeroth-order optimization literature (See Appendix C). The non-differentiability of Equation (2) renders the entire optimization problem non-convex. The convergence is evaluated using the first-order stationary condition via the squared gradient norm (Proof in Appendix D):

$$\frac{1}{\mathcal{T}}\sum_{t=1}^{\mathcal{T}}\mathbb{E}\left[\|g(u)\|_2^2\right] = O(\delta^2 + \frac{1}{b}),\tag{11}$$

where $\mathcal{T}$ denotes the update iteration. The convergence upper bound is

$$\mathbb{E}[g(u_{\mathcal{T}}) - g(u^*)] \leq O(\frac{1}{\sqrt{\mathcal{T}}}).\tag{12}$$

Moreover, more details on analysis of sources of error in ZOO are provided in Appendix E.

**Loss function.** The total loss function, incorporating both the temporal error term (TET) (Deng et al., 2022) and the MSE loss, given by

$$\mathcal{L} = (1-\lambda)\frac{1}{T}\sum_{t=1}^{T}\mathcal{L}_{\text{CE}}(u_t, \mathbf{y}) + \lambda\mathcal{L}_{\text{MSE}}(u_t, \mathbf{y}),\tag{13}$$

where $T$ denotes the number of time steps, and $\lambda$ is a hyperparameter that governs the relative contribution of the temporal prediction error term and $\mathcal{L}_{\text{MSE}}$.

## 5.5 FAST ADAPTIVE PROTOTYPE SUBSPACE PROJECTION

Class prototypes averaging extracted features often cause discrepancies with actual data distributions. Prototypes learned from the scarce and narrow-distribution samples of novel classes are prone to bias in FSCIL (Liu et al., 2020a). To mitigate this bias, we introduce prototype orthogonal subspace projection, which updates novel-class prototypes in two steps: (i) projecting novel-class prototypes onto the orientation subspace spanned by base-class prototypes; and (ii) blending the projected prototype with the original novel prototype through a convex combination. As denoted in Section 3.2, we further define two specific prototypes that belong to $\mathcal{C}_B$ base classes and $\mathcal{C}_N$ new classes as $\tilde{\mathbf{B}} = \left[\frac{\mathbf{P}_1^b}{\|\mathbf{P}_1^b\|_2}, \frac{\mathbf{P}_2^b}{\|\mathbf{P}_2^b\|_2}, \dots, \frac{\mathbf{P}_B^b}{\|\mathbf{P}_B^b\|_2}\right]^\top \in \mathbb{R}^{B \times D}$ and $\tilde{\mathbf{C}} = \left[\frac{\mathbf{P}_1^n}{\|\mathbf{P}_1^n\|_2}, \frac{\mathbf{P}_2^n}{\|\mathbf{P}_2^n\|_2}, \dots, \frac{\mathbf{P}_N^n}{\|\mathbf{P}_N^n\|_2}\right]^\top \in \mathbb{R}^{N \times D}$, respectively, where $D$ is the feature dimension, $B = |\mathcal{C}_B|$, and $N = |\mathcal{C}_N|$. To obtain information on the subspace structure spanned by the base classes, we construct a new prototype representation $\mathbf{G}$ to represent a generalized inverse of a covariance-like matrix in projection calculation, given by

$$\mathbf{G} = \tilde{\mathbf{B}}(\tilde{\mathbf{B}}^\top\tilde{\mathbf{B}})^{-1}\tilde{\mathbf{B}}^\top.\tag{14}$$

Note that the normalization term $\tilde{\mathbf{B}}^\top\tilde{\mathbf{B}}$ is necessary because the generation process of the subspace basis is not guaranteed to be orthogonal to each other. Hence, we construct $|\mathcal{C}_B|$ distinct orthogonal subspaces represented by $\mathbf{G}$ for base classes. As classes increase, diverse class information allows extending classifiers with prototypes. The projection vectors $\tilde{\mathbf{P}}_{\text{proj}}$ are updated by mapping the coordinates in the base subspace back to the original $D$-dimensional space, denoted as $\tilde{\mathbf{P}}_{\text{proj}} = \tilde{\mathbf{C}}\mathbf{G}$. Finally, we integrate base and new knowledge by reconstructing the new information within the base subspace by

$$\tilde{\mathbf{P}} = (1-\alpha)\tilde{\mathbf{C}} + \alpha\tilde{\mathbf{P}}_{\text{proj}},\tag{15}$$

where the trade-off coefficient $\alpha$ further balances the integration of base and novel knowledge. The current classifier weights are updated by $\tilde{\mathbf{P}}$ to enable new class prediction (See Appendix F for more analysis). The orthogonal projection measures the alignment between novel and base classes, where a larger projection magnitude implies stronger cosine similarity. Consequently, features with higher similarity contribute more to the prototype update. Compared to the raw prototype $\tilde{C}$, the updated prototype lies closer to the expected semantic direction and captures more discriminative features.

Table 1: Comparison with SOTA methods on MiniImageNet dataset for On-Device FSCIL.

| Method | Accuracy in each session (%) ↑ | | | | | | | | | $\mathcal{A}_{\text{avg}}$ ↑ | $\Delta_{\text{last}}$ |
| --- | --- | --- | --- | --- | --- | --- | --- | --- | --- | --- | --- |
| | 0 | 1 | 2 | 3 | 4 | 5 | 6 | 7 | 8 | | |
| CEC (SNN) | $54.33_{\pm 2.14}$ | $49.81_{\pm 1.98}$ | $46.82_{\pm 1.82}$ | $44.28_{\pm 1.69}$ | $42.28_{\pm 1.63}$ | $40.31_{\pm 1.40}$ | $38.26_{\pm 1.36}$ | $36.75_{\pm 1.33}$ | $35.78_{\pm 1.29}$ | $43.18_{\pm 1.63}$ | +12.92 |
| FACT (SNN) | $63.03_{\pm 0.91}$ | $52.51_{\pm 1.52}$ | $49.25_{\pm 1.28}$ | $46.59_{\pm 1.13}$ | $44.32_{\pm 1.19}$ | $42.35_{\pm 1.24}$ | $40.16_{\pm 1.17}$ | $38.47_{\pm 1.08}$ | $37.44_{\pm 1.17}$ | $46.01_{\pm 1.18}$ | +11.26 |
| S3C (SNN) | $31.55_{\pm 0.51}$ | $16.52_{\pm 0.97}$ | $17.89_{\pm 0.84}$ | $31.17_{\pm 0.97}$ | $24.69_{\pm 0.82}$ | $29.02_{\pm 0.86}$ | $15.35_{\pm 0.62}$ | $27.30_{\pm 0.22}$ | $26.56_{\pm 0.04}$ | $24.45_{\pm 0.19}$ | +22.14 |
| BIDIST (SNN) | $43.72_{\pm 0.32}$ | $40.66_{\pm 0.45}$ | $37.81_{\pm 0.69}$ | $35.51_{\pm 0.79}$ | $33.58_{\pm 0.61}$ | $31.72_{\pm 0.57}$ | $30.12_{\pm 0.64}$ | $28.95_{\pm 0.70}$ | $27.85_{\pm 0.49}$ | $34.44_{\pm 0.58}$ | +20.85 |
| SAVC (SNN) | $41.75_{\pm 0.41}$ | $38.54_{\pm 0.30}$ | $35.79_{\pm 0.21}$ | $33.40_{\pm 0.13}$ | $31.31_{\pm 0.06}$ | $29.47_{\pm 1.00}$ | $27.83_{\pm 0.94}$ | $26.37_{\pm 0.89}$ | $25.05_{\pm 0.85}$ | $32.17_{\pm 1.08}$ | +23.65 |
| TEEN (SNN) | $62.87_{\pm 1.69}$ | $52.75_{\pm 1.15}$ | $49.64_{\pm 1.01}$ | $46.91_{\pm 0.85}$ | $44.82_{\pm 0.85}$ | $42.81_{\pm 0.78}$ | $40.59_{\pm 0.77}$ | $39.18_{\pm 0.85}$ | $38.01_{\pm 0.84}$ | $46.40_{\pm 0.96}$ | +10.69 |
| WARP (SNN) | $50.07_{\pm 4.16}$ | $31.67_{\pm 3.72}$ | $29.93_{\pm 3.50}$ | $28.07_{\pm 3.41}$ | $25.49_{\pm 4.23}$ | $24.28_{\pm 4.32}$ | $22.84_{\pm 4.34}$ | $22.13_{\pm 4.16}$ | $21.30_{\pm 3.96}$ | $27.47_{\pm 3.71}$ | +27.40 |
| CLOSER (SNN) | $65.88_{\pm 0.09}$ | $61.39_{\pm 0.08}$ | $57.72_{\pm 0.03}$ | $54.74_{\pm 0.10}$ | $52.34_{\pm 0.07}$ | $49.98_{\pm 0.21}$ | $47.69_{\pm 0.18}$ | $45.94_{\pm 0.14}$ | $44.69_{\pm 0.07}$ | $53.38_{\pm 0.09}$ | +4.01 |
| ALADE (FSCIL) | $57.91_{\pm 0.45}$ | $47.04_{\pm 0.62}$ | $44.07_{\pm 0.55}$ | $41.75_{\pm 0.45}$ | $39.72_{\pm 0.50}$ | $37.92_{\pm 0.47}$ | $36.04_{\pm 0.48}$ | $34.57_{\pm 0.45}$ | $33.73_{\pm 0.49}$ | $41.42_{\pm 0.47}$ | +14.97 |
| **SAFA-SNN** | $\mathbf{74.66}_{\pm 0.62}$ | $\mathbf{68.93}_{\pm 0.46}$ | $\mathbf{64.62}_{\pm 0.57}$ | $\mathbf{61.29}_{\pm 0.88}$ | $\mathbf{58.30}_{\pm 0.95}$ | $\mathbf{55.38}_{\pm 0.78}$ | $\mathbf{52.60}_{\pm 0.62}$ | $\mathbf{50.58}_{\pm 0.60}$ | $\mathbf{48.70}_{\pm 0.67}$ | $\mathbf{59.45}_{\pm 0.67}$ | |

## 6 EXPERIMENTS

### 6.1 EXPERIMENTS SETUP

**Datasets and Spiking architecture.** We evaluate the generalization performance of SAFA-SNN on two standard benchmark datasets, i.e., CIFAR-100 (Krizhevsky et al., 2009) and MiniImageNet (Russakovsky et al., 2015), each split into eight 5-way 5-shot incremental tasks. We also extend experiments on three neuromorphic datasets: CIFAR10-DVS (Li et al., 2017), DVS128 Gesture (Amir et al., 2017), and N-Caltech101 (Orchard et al., 2015). CIFAR10-DVS is split into four 1-way 1-shot tasks,

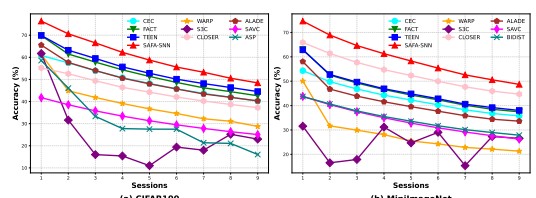

Figure 3: Accuracy curves in each session.

DVS128 Gesture into five 1-way 1-shot tasks, and N-Caltech101 into eight 5-way 5-shot incremental tasks. We adopt Spiking VGG variants (5, 9, 11), Spiking Resnet variants (18, 19, 20), and Spikingformer. Further details can be found in Appendix G.

**Realistic Implementation.** All experiments are conducted on a mobile platform NVIDIA Jetson AGX Orin (NVIDIA, 2025), which features a 12-core Arm Cortex-A78AE processor, supporting 64-bit Armv8.2 architecture, with 60W peak power and idle power below 15W.

**Training Details.** Our implementation uses PyTorch with Adam optimizer. Models are trained for 300 epochs on CIFAR-100 and MiniImageNet with a batch size of 128, and for 100 epochs on CIFAR10-DVS with a batch size of 32, on DVS128 Gesture and N-Caltech101 with a smaller batch size of 8. The learning rate, $\beta$, $\theta$, $b$, and $\delta$ are set to 0.001, 1.2, 0.001, 5, and 0.5, respectively. Each experiment is repeated three times with different seeds.

### 6.2 COMPARISON WITH STATE-OF-THE-ART METHODS

**Baselines.** We select nine FSCIL methods in their SNN version, i.e., CEC (Zhang et al., 2021), FACT (Zhou et al., 2022), S3C (Kalla & Biswas, 2022), BIDIST (Zhao et al., 2023), SAVC (Song et al., 2023), TEEN (Wang et al., 2023), WARP (Kim et al., 2023), CLOSER (Oh et al., 2024), and FSCIL-ASP (Liu et al., 2024a), as baselines. We adopt an SNN-based CIL method, ALADE-SNN (Ni et al., 2025), and two SNN training methods (Kim et al., 2022; Meng et al., 2023) to evaluate FSCIL performance. See Appendix H for details.

Table 2: Comparative results on neuromorphic datasets.

| Dataset | Method | $\mathcal{A}_{\text{avg}}(\%)$ | $\mathcal{A}_{\text{last}}(\%)$ |
| --- | --- | --- | --- |
| CIFAR10-DVS | WARP | $30.92_{\pm 0.70}$ | $12.75_{\pm 0.14}$ |
| | TEEN | $40.57_{\pm 1.02}$ | $34.60_{\pm 0.79}$ |
| | **SAFA-SNN** | $\mathbf{47.56}_{\pm 0.94}$ | $\mathbf{36.96}_{\pm 0.38}$ |
| DVS128 Gesture | WARP | $35.41_{\pm 1.76}$ | $13.02_{\pm 1.25}$ |
| | TEEN | $83.13_{\pm 0.80}$ | $74.31_{\pm 1.42}$ |
| | **SAFA-SNN** | $\mathbf{86.74}_{\pm 0.69}$ | $\mathbf{77.91}_{\pm 0.63}$ |
| N-Caltech101 | WARP | $25.34_{\pm 1.63}$ | $14.40_{\pm 2.07}$ |
| | TEEN | $34.23_{\pm 0.18}$ | $27.86_{\pm 0.49}$ |
| | **SAFA-SNN** | $\mathbf{45.68}_{\pm 0.63}$ | $\mathbf{39.69}_{\pm 0.67}$ |

Table 3: Comparative results with SNN-based training methods for on-device FSCIL.

| Method | Dataset | Time Step | Backbone | Param Size (M) | $\mathcal{A}_{\mathrm{h}}(\%)$ | $\mathcal{A}_{\mathrm{last}}(\%)$ | $\mathcal{A}_{\mathrm{avg}}(\%)$ |
|---|---|---|---|---|---|---|---|
| Early Bird | CIFAR-100 | 4 | Spiking VGG-5 | 221.35 | $24.20_{\pm 0.54}$ | $41.21_{\pm 0.66}$ | $50.87_{\pm 0.60}$ |
| (Kim et al., 2022) | CIFAR-100 | 5 | Spiking VGG-9 | 106.87 | $22.25_{\pm 0.85}$ | $39.92_{\pm 1.07}$ | $49.30_{\pm 1.49}$ |
| SLTT | CIFAR-100 | 6 | Spiking VGG-5 | 32.85 | $25.17_{\pm 1.05}$ | $41.63_{\pm 0.27}$ | $51.12_{\pm 0.28}$ |
| (Meng et al., 2023) | Mini-ImageNet | 4 | Spiking VGG-9 | 106.87 | $18.93_{\pm 0.31}$ | $32.58_{\pm 0.77}$ | $40.45_{\pm 0.64}$ |
| | CIFAR10-DVS | 4 | Spiking VGG-9 | 262.87 | $10.87_{\pm 0.62}$ | $26.60_{\pm 1.27}$ | $34.46_{\pm 0.85}$ |
| SAFA-SNN | CIFAR-100 | 4 | Spiking VGG-5 | 32.86 | $\mathbf{27.32}_{\pm 0.91}$ | $\mathbf{46.47}_{\pm 0.53}$ | $\mathbf{56.72}_{\pm 0.25}$ |
| | Mini-ImageNet | 4 | Spiking VGG-9 | 22.63 | $\mathbf{24.67}_{\pm 0.55}$ | $\mathbf{49.25}_{\pm 0.24}$ | $\mathbf{60.98}_{\pm 0.62}$ |
| | CIFAR10-DVS | 4 | Spiking VGG-9 | 262.63 | $\mathbf{14.47}_{\pm 0.86}$ | $\mathbf{35.65}_{\pm 0.48}$ | $\mathbf{41.67}_{\pm 0.34}$ |

**Accuracy.** We test the Top-1 accuracy on all seen tasks $0, 1, ...s$ after the session $s$. $\Delta_{\mathrm{last}}$ means our relative improvement in the last session. Harmonic accuracy measures the balance between base and novel class performance after each session $s$, i.e., $\mathcal{A}_{\mathrm{h}} = \frac{2 \times \mathcal{A}_{\mathrm{b}} \times \mathcal{A}_{\mathrm{n}}}{\mathcal{A}_{\mathrm{b}} + \mathcal{A}_{\mathrm{n}}}$, where $\mathcal{A}_{\mathrm{b}}$ denotes test accuracy in base session, and $\mathcal{A}_{\mathrm{n}}$ the average accuracy over sessions $s > 0$. As reported in Table 1, SAFA-SNN surpasses the second-best approach by 4.01% for improvement in the final session and boosts the average performance by 6.07% on Mini-ImageNet. The performance curves presented in Figure 3 show that SAFA-SNN achieves state-of-the-art performance across CIFAR-100 and Mini-ImageNet, respectively. Table 2 shows accuracy on average and the last session compared with WARP and TEEN on three neuromorphic datasets, verifying its generalization and robustness performance. We find baseline WARP has the worst performance, indicating that the ability of space compaction to obtain effective parameter presentation is totally limited, far inferior to our neuronal dynamics and subspace projection. Table 3 shows that our method outperforms SNN training methods in different settings. This confirms the effectiveness and potential of our method on the SNN deployment. Additional results on more models, comparison with soft subnetwork-based FSCIL methods, and metric $\mathcal{A}_{\mathrm{h}}$ are provided in Appendix I.1, I.2, and I.3 respectively.

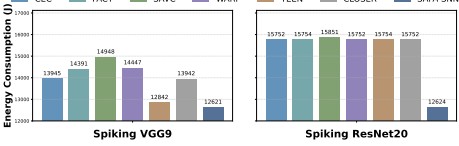

Figure 4: Actual energy cost in training.

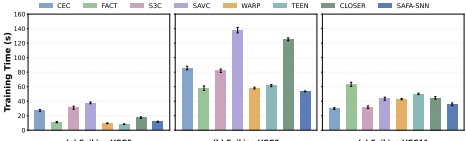

Figure 5: Average training time.

**Energy consumption.** The measurement of *actual energy consumption* in training uses the built-in sensor values (e.g., GPU, I/O) between the RAM and storage on the Jetson Orin AGX device (NVIDIA, 2025), which is gauged by multiplying power (W) by time. We put the training energy consumption in Figure 4. It can be seen that SAFA-SNN exhibits notably lower energy consumption than baselines. Following Yao et al. (2023), we present the *theoretical inference energy consumption* of different ANNs and SNNs in Appendix J, with detailed formulas and results.

**Time efficiency.** Figure 5 compares the time efficiency of SAFA-SNN and baselines with Spiking VGGs. Our neuronal dynamics, combined with ZOO and prototype subspace projection, reduce unnecessary training time without modifying synaptic weights or adjusting parameter spaces, outperforming baselines in both efficiency and effectiveness. (Described in detail in Appendix K).

## 6.3 ABLATION STUDY

**Ablation on SAFA-SNN.** We conduct an ablation study to analyze the importance of each component in SAFA-SNN: Sparsity-Aware neuronal dynamics (SA), Zeroth-Order Optimization (ZOO), and subspace projection of prototypes (SP). We report incremental performance curves on CIFAR-100 and MiniImageNet with time step 4 and Spiking VGG-9, as shown in Figure 6. We can infer that SA has the worst performance, since it does not consider possible adjustments in feature space, and we view it as the baseline. When equipped with SP, it shows significant performance gains as the model adapts to extract more informative features from base prototypes. We then use ZOO for gradient estimation, as shown in the highest curve, which corresponds to our full SAFA-SNN. Ablations verify that every component in SAFA-SNN boosts FSCIL performance.

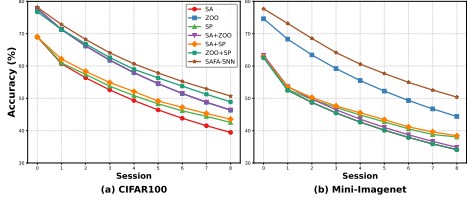 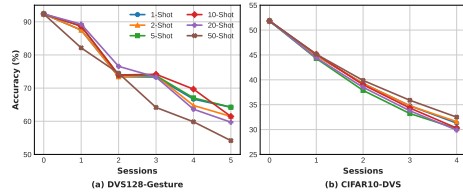

Figure 6: Ablation results of SAFA-SNN.  Figure 7: All-session accuracy of variant shots.

**Effects on different N-way-K-shots.** We assess shot count impact on accuracy by varying 1,2,5,10,15,20,50 on DVS128 Gesture and CIFAR10-DVS in Figure 7 (Analysis in Appendix L).

**Hyper-parameters Analysis.** Results on the effects of key parameters (i.e., $\beta$, $\gamma$, $b$, $\delta$, $\eta$ and $\lambda$) and time step $T$, are provided in the Appendix M and N, respectively.

**Sparsity-Accuracy trade-off Analysis.** The sparsity remains up to 80% even when setting $T = 2, 3, 4$ on different datasets through the training process, indicating a balance in spiking sparsity and accuracy, showing potential computation efficiency (Results in Appendix O).

### 6.4 MORE COMPARISON ANALYSIS

**Comparison with surrogate gradients.** ZOO is a class of gradient-free optimization methods that approximate true gradients using only function evaluations. Unlike surrogate gradients, which replace the non-differentiable spike gradient with a limited-width approximation, ZOO addresses a stochastic optimization problem for the non-differentiable $\frac{\partial \mathbf{S}_t}{\partial \mathbf{U}_t}$ in a non-convex setting. We compare the performance of ZOO with six surrogate gradients, with results and analysis in Appendix P.1.

**Comparison with projection-based methods.** To achieve a comprehensive evaluation, in this part, we present comparisons with typical projection-based methods using ResNet backbones in Appendix P.2. Specifically, the comparison methods include WARP (Kim et al., 2023) and Subspace-reg (Akyürek et al., 2022). The results show that SAFA-SNN consistently outperforms other subspace regulation methods in most cases.

**Comparison results with ANN-based PEFT methods for FSCIL.** While ANN-based FSCIL methods can achieve slightly higher accuracy, SNNs remain highly competitive and provide significant advantages in efficiency and deployability. The observed performance gap is primarily due to the fact that the PEFT backbones of ANN methods are pretrained on ImageNet-1K, whereas our lightweight, spike-driven transformer is specifically designed for on-device FSCIL. In Appendix P.3, we compare against three categories of PEFT-based recent methods: ASP (Liu et al., 2024a), CoACT (Roy et al., 2024), and PriViLege (Park et al., 2024).

## 7 DISCUSSION

A potential limitation is the performance degradation in deeper networks, such as Spiking ResNet-34, caused by the failure of identity mapping. This limitation can be mitigated by employing SEW-style residual connections. In addition, assuming a fixed number of classes per session oversimplifies the dynamics of real-world data streams. Future work will address imbalanced way-shot settings to better reflect practical on-device FSCIL. Analysis on base and new classes, visualization, and further discussion are provided in Appendix Q, R, and S.

## 8 CONCLUSION

This paper focuses on few-shot class-incremental learning with SNN for on-device scenarios. We proposed SAFA-SNN by incorporating sparsity-aware neuronal dynamics, zeroth-order optimization for non-differential spike, and subspace projection for updating new-class prototypes. Extensive experiments on benchmark and neuromorphic datasets indicate that SAFA-SNN outperforms existing FSCIL methods in both performance and energy efficiency with realistic implementation.

ACKNOWLEDGMENTS

The work of this paper is supported by the National Natural Science Foundation of China under Grant No. 62502443, No. 62125206, and the National Key Research and Development Program of China under Grant No. 2025YFG0100700, No. 2022YFB4500100.

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

## A  SPARSITY-AWARE NEURONAL DYNAMICS

First, active neurons facilitate knowledge updating while stable neurons align with base-class representations to preserve learned knowledge. This creates a clear distinction between neurons encoding base classes and those encoding novel classes, as base classes are abundant while novel classes are scarce in FSCIL. It implements a method similar to subnet regularization, but unlike soft subnetwork approaches (Mazumder et al., 2021; Kang et al., 2023; 2024), which update a task-specific subnet for each task, incurring additional computational overhead and introducing extra trainable parameters. Adaptive neurons are more plastic and can adapt their firing thresholds more rapidly to encode new-class information. The whole neuronal dynamics can be formulated as:

$$\mathbf{I}_t = \text{BN}\big(\text{Conv}(\mathbf{X}_t; \boldsymbol{\theta})\big), \tag{16}$$

$$\mathbf{U}_t = \tau \mathbf{U}_{t-1} + \mathbf{I}_t, \tag{17}$$

$$\mathbf{S}_t = H(\mathbf{U}_t - \mathbf{U}_{\text{th}}), \tag{18}$$

$$\mathbf{A} = \beta(\mathbf{1} - \mathbf{M}) + \gamma \mathbf{M}, \tag{19}$$

$$\mathbf{U}'_{\text{th}} = \mathbf{U}_{\text{th}} + \mathbf{A}\big(\mathbf{r}_c - \mathbf{r}_b\big). \tag{20}$$

Initially, we divide neurons into act as active and stable neurons with a mask $\mathbf{M}$. At time step $t$, $\mathbf{I}_t$ is the cumulative current with the input $\mathbf{X}_t$ and parameters $\theta$. The membrane potential $\mathbf{U}_t$ evolves continuously from time step $t - 1$ to time step $t$. When the membrane potential rises and reaches threshold $\mathbf{U}_{\text{th}}$, it emits a spike $\mathbf{S}_t$. Then $\mathbf{U}_{\text{th}}$ changes by variation of sparsity with a regulator $\mathbf{A}$. Sparsity-aware neuronal dynamics ensures that most neurons exhibit minimal threshold changes and maintain their spike firing within a constrained range under varying sparsity, resulting in relatively stable spike representations of base-class knowledge. Meanwhile, a small subset of neurons with a broader range of dynamic threshold adaptation in response to sparsity minimizes updates to the majority of synapses, improves incremental learning adaptability, and mitigates catastrophic forgetting.

## B  MEAN SQUARE APPROXIMATION ERROR ANALYSIS

The mean square approximation error (MSE) of the gradient estimate $\hat{\nabla}g(x)$ with respect to $\nabla g(x)$:

$$\mathbb{E}[\|\hat{\nabla}g(x) - \nabla g(x)\|_2^2] = O(\delta^2) + O(\frac{d}{b\delta^2}), \tag{21}$$

where we use the big $O$ notation to emphasize the dominant factors, i.e., the dimensionality $d$ and the perturbation radius (or smoothing parameter) $\delta$, and the number of samples $b$ that govern the gradient estimation error (Berahas et al., 2022). At the non-differentiable point $u = 0$, the true gradient does not exist. However, when $z$ is drawn from a symmetric distribution, the estimator is unbiased, i.e., its expectation is zero. Importantly, the non-zero contributions of the estimator are concentrated in a neighborhood around $u = 0$, where the function exhibits non-smooth behavior.

The parameter $\delta$ plays a crucial role in gradient estimation (Liu et al., 2020b). A smaller $\delta$ reduces the bias of the estimator but significantly increases its variance, as the finite-difference computation is more sensitive to noise. In practice, an excessively small $\delta$ may cause the function difference to be dominated by stochastic noise, making the estimator unreliable. In contrast, a larger $\delta$ reduces variance through increased smoothing but introduces greater bias.

As the sample size $b$ increases, the estimator converges to its expectation according to the law of large numbers. In the limit as $b \to \infty$, the estimator converges to the gradient of the smoothed objective function rather than the true gradient of the original, non-smoothed objective. At differentiable points, the bias diminishes as $\delta \to 0$, while the variance of a single-point estimation is of order $O(\frac{1}{\delta^2})$. Through multi-point averaging, the variance is reduced to $O(\frac{1}{b\delta^2})$, indicating that increasing $b$ can effectively compensate for the variance amplification caused by small $\delta$.

## C  CRITERIA FOR CONVERGENCE ANALYSIS

We provide four assumptions regarding convergence rates on zeroth-order algorithms, which can be applicable to different types of problems according to the previous literature (Liu et al., 2020b).

*(1) Convex optimization*: For a convex function $f$, convergence is evaluated by

$$\mathbb{E}[f(x_{\mathcal{T}}) - f(x^*)],\tag{22}$$

where $x_{\mathcal{T}}$ denotes the point obtained at the final iteration $\mathcal{T}$ and $x^* \in \underset{x \in \mathcal{X}}{\operatorname{argmin}} f(x)$ denotes an optimal solution of $f$, with $\mathcal{X}$ as a closed convex set. The expectation accounts for all randomness.

*(2) Online convex optimization*: In the online setting, performance is measured by the cumulative regret $\operatorname{Regret}_{\mathcal{T}}$ (Hazan et al., 2016) for a sequence of convex cost functions $\{f_t\}_{t=1}^{\mathcal{T}}$ as

$$\mathbb{E}[\sum_{t=1}^{\mathcal{T}} f_t(x_t) - \min_{x \in \mathcal{X}} \sum_{t=1}^{\mathcal{T}} f_t(x)].\tag{23}$$

*(3) Unconstrained nonconvex optimization*: In unconstrained nonconvex settings, convergence is assessed by the average expected squared gradient norm across $\mathcal{T}$ iterations:

$$\frac{1}{\mathcal{T}} \sum_{t=1}^{\mathcal{T}} \mathbb{E}\left[\|\nabla f(x_t)\|_2^2\right].\tag{24}$$

*(4) Constrained nonconvex optimization*: For optimization over a feasible set $X$, convergence is often measured by the norm of the projected gradient (Ghadimi et al., 2016; J Reddi et al., 2016):

$$P_X(x_t, \nabla f(x_t), \eta_t) := \frac{1}{\eta_t}[x_t - \Pi_X(x_t - \eta_t \nabla f(x_t))],\tag{25}$$

where $\Pi_X$ denotes the projection operator that updates a point subject to the constraint $x \in \mathcal{X}$, and $\|P_X\|$ measures how far the current iterate $x_t$ is from satisfying first-order optimality within $X$.

Overall, ZOO is typically applied to *unconstrained nonconvex optimization* because it does not require gradient information and can handle highly complex, nonconvex functions. Introducing constraints would complicate the gradient estimation and increase bias and variance.

## D  PROOF OF ZEROTH-ORDER OPTIMIZATION

We provide a theoretical proof of the estimation function $g(x)$ according to the literature (Mukhoty et al., 2023) as follows.

**Definition 1.** *A function $g : \mathbb{R} \to \mathbb{R}_{\geq 0}$ is called a* surrogate function *if it is even, non-decreasing on $(-\infty, 0)$, and $c := \int_{-\infty}^{\infty} g(z)\, dz < \infty$.*

**Lemma 1.** *Let $\lambda$ be a distribution such that*

$$\int_0^{\infty} z^{\alpha+1} \lambda(z)\, dz < \infty.\tag{26}$$

*Then the expectation $\mathbb{E}_{z \sim \lambda}[g^2(u; z, \delta)]$ defines a surrogate function.*

**Theorem 1.** *Let $p$ be a probability distribution with probability density function (PDF) $p(t)$ such that $\int_0^{\infty} t^{\alpha} p(t)dt < \infty$ and $\int_0^{\infty} t^{\alpha+1} p(t)dt < \infty$. Define the transformed distribution $\tilde{\lambda}$ with PDF*

$$\tilde{\lambda}(z) = \frac{1}{c} \int_{|z|}^{\infty} t^{\alpha} \lambda(t)\, dt,\tag{27}$$

*where $c$ is a normalization constant ensuring $\int_{-\infty}^{\infty} \tilde{\lambda}(z)dz = 1$. Then*

$$\mathbb{E}_{z \sim p}[g^2(u; z, \delta)] = \frac{d}{du} \mathbb{E}_{z \sim \tilde{\lambda}}[h(u + \delta z)],\tag{28}$$

*where $h$ is an appropriately defined surrogate function.*

The standard normal PDF is $\frac{1}{\sqrt{2\pi}} \exp\left(-\frac{z^2}{2}\right)$, so that

$$\mathbb{E}_{z \sim p}[g^2(u; z, \delta)] = \int_{-\infty}^{\infty} \frac{|z|}{2\delta} \frac{1}{\sqrt{2\pi}} \exp\left(-\frac{z^2}{2}\right) dz = \frac{1}{\delta\sqrt{2\pi}} \exp\left(-\frac{u^2}{2\delta^2}\right).\tag{29}$$

**Theorem 2.** *Let $h(u)$ be a surrogate function and define*

$$c := -2\delta^2 \int_0^\infty \frac{1}{z^\alpha} h'(z\delta) dz < \infty, \quad p(z) := -\frac{\delta^2}{cz^\alpha} h'(z\delta), \; z > 0. \tag{30}$$

*Then $p(z)$ is a valid PDF and*

$$c\, \mathbb{E}_{z\sim p}[g^2(u; z, \delta)] = h(u), \tag{31}$$

*with derivative*

$$h'(u) = -\frac{k^2 \exp(-ku)(1 - \exp(-ku))}{(1 + \exp(-ku))^3}. \tag{32}$$

Consider the Sigmoid surrogate function, where the differentiable Sigmoid (Zenke & Ganguli, 2018) approximates the Heaviside function. The corresponding surrogate gradient is defined as

$$h(u) := \frac{d}{du} \frac{1}{1 + \exp(-ku)} = \frac{k \exp(-ku)}{(1 + \exp(-ku))^2}. \tag{33}$$

It is easy to verify that $h(u)$ satisfies the properties of a surrogate function: it is even, non-decreasing on $(-\infty, 0)$, and integrable over $\mathbb{R}$ with $\int_{-\infty}^\infty h(u)\, du = 1 < \infty$.

Applying Theorem 2, the scaling constant $c$ can be computed as

$$c = -2\delta^2 \int_0^\infty \frac{h'(t\delta)}{t}\, dt = 2\delta^2 k^2 \int_0^\infty \frac{\exp(-k\delta t)(1 - \exp(-k\delta t))}{t(1 + \exp(-k\delta t))^3}\, dt = \frac{\delta^2 k^2}{a^2}, \tag{34}$$

where $a := \sqrt{\frac{1}{0.4262}}$. The corresponding probability density function is

$$p(z) = -\frac{\delta^2}{c} \frac{h'(\delta z)}{z} = \frac{a^2 \exp(-k\delta z)(1 - \exp(-k\delta z))}{z(1 + \exp(-k\delta z))^3}, \quad z > 0. \tag{35}$$

## E  SOURCES OF ERROR FOR ZEROTH-ORDER OPTIMIZATION

We employ a two-point ZOO estimator, which queries the model output under small symmetric perturbations and yields a significantly more accurate gradient estimate than one-point approximations (Liu et al., 2020b).

We first analyze the gradient approximation error of ZOO, denoted by $\|g(x) - \nabla\Phi(x)\|$, where $\Phi(x)$ is the true objective function and $\nabla\Phi(x)$ its exact gradient. According to Berahas et al. (2022), the sources of error in approximating the gradient can be summarized into two main folds:

(1) The discrepancy between the true gradient $\nabla\Phi(x)$ and the gradient of the Gaussian-smoothed surrogate $F(x)$, which is constructed based on the noisy observations of $f$;

(2) The error introduced when estimating $\nabla F(x)$ through a finite number of sample averages.

Derivative-free optimization often proceeds by approximating the true gradient $\nabla\Phi(x)$ with an estimate $g(x)$ calculated from noisy function evaluations, which is then used in a gradient-based update. By adding and subtracting $\nabla F(x)$, the gradient approximation error can be decomposed as

$$\begin{aligned} \|g(x) - \nabla\Phi(x)\| &= \|(\nabla F(x) - \nabla\Phi(x)) + (g(x) - \nabla F(x))\| \\ &\leq \|\nabla F(x) - \nabla\Phi(x)\| + \|g(x) - \nabla F(x)\|. \end{aligned} \tag{36}$$

Bound on the first term $\|\nabla F(x) - \nabla\Phi(x)\|$. Assume that the noise in the function evaluations $\epsilon(x)$ is bounded for all $x \in \mathbb{R}^n$, and that $\Phi$ has $L$-Lipschitz continuous gradients. Suppose the function $f$ is an approximation of $\Phi$ with uniformly bounded error, i.e., there exists $\epsilon_f \geq 0$ such that

$$|f(x) - \Phi(x)| = |\epsilon(x)| \leq \epsilon_f, \quad \forall x \in \mathbb{R}^n. \tag{37}$$

If $\Phi$ has $L$-Lipschitz continuous gradients, then

$$\|\nabla F(x) - \nabla\Phi(x)\| \leq \sqrt{b}\, L\delta + \frac{\sqrt{b}\, \epsilon_f}{\delta}, \tag{38}$$

where $\delta$ is the sampling radius and $b$ is the sample number. If $\Phi$ has $M$-Lipschitz continuous Hessians, then

$$\|\nabla F(x) - \nabla \Phi(x)\| \leq nM\delta^2 + \frac{\sqrt{b}\,\epsilon_f}{\delta}. \quad (39)$$

Bound on the second term $\|g(x) - \nabla F(x)\|$. This error arises from the sample-average approximation. Assume that $\Phi$ has $L$-Lipschitz continuous gradients and its Hessian is $M$-Lipschitz continuous. Then the Gaussian-smoothed function $F$ inherits the corresponding smoothness properties. Let $g(x)$ denote the two-point estimator computed using $N$ independent Gaussian directions $u_i \sim \mathcal{N}(0, I)$. The covariance of $g(x)$ satisfies

$$\text{Var}(g(x)) = \frac{1}{N} \mathbb{E}_{u \sim \mathcal{N}(0,I)} \left[ \left( \frac{f(x + \delta u) - f(x - \delta u)}{2\delta} \right)^2 uu^T \right] - \frac{1}{N} \nabla F(x) \nabla F(x)^T. \quad (40)$$

Using the result from Berahas et al. (2022), when $g(x)$ is computed using the two-point formula, for all $x \in \mathbb{R}^n$, $\text{Var}(g(x)) \preceq \kappa(x)I$, where

$$\kappa(x) = \frac{3}{N} \left( 3\|\nabla \Phi(x)\|^2 + \frac{M^2\delta^4}{36}(n + 2)(n + 4)(n + 6) + \frac{\epsilon_f^2}{\delta^2} \right). \quad (41)$$

Table 4: Comparison with FSCIL baselines on static dataset CIFAR-100 with 5way-5shot incremental learning setting and Spiking VGG-9.

| Method | Accuracy in each session (%) ↑ | | | | | | | | | $\mathcal{A}_{\text{avg}}$ ↑ | $\Delta_{\text{last}}$ |
|---|---|---|---|---|---|---|---|---|---|---|---|
| | 0 | 1 | 2 | 3 | 4 | 5 | 6 | 7 | 8 | | |
| CEC | 60.82 | 57.49 | 53.93 | 50.80 | 48.18 | 45.75 | 43.69 | 41.95 | 40.22 | 49.20 | +7.87 |
| FACT | 69.82 | 61.46 | 57.83 | 54.33 | 51.35 | 48.64 | 46.20 | 44.45 | 42.61 | 52.97 | +5.48 |
| S3C | 61.72 | 31.60 | 16.00 | 15.40 | 11.00 | 19.40 | 18.00 | 25.20 | 23.00 | 24.59 | +25.09 |
| BIDIST | 61.30 | 57.69 | 54.17 | 50.88 | 48.33 | 45.72 | 43.82 | 42.08 | 40.36 | 49.37 | +7.73 |
| SAVC | 41.75 | 38.54 | 35.79 | 33.40 | 31.31 | 29.47 | 27.83 | 26.37 | 25.05 | 32.17 | +23.04 |
| TEEN | 69.87 | 63.20 | 59.49 | 55.67 | 52.79 | 50.11 | 48.12 | 46.33 | 44.51 | 59.34 | +3.58 |
| WARP | 61.32 | 44.88 | 41.84 | 39.11 | 36.68 | 34.60 | 32.20 | 31.07 | 28.74 | 38.94 | +19.35 |
| CLOSER | 55.22 | 52.55 | 49.39 | 46.41 | 44.23 | 42.06 | 40.23 | 38.67 | 37.21 | 45.10 | +10.88 |
| ASP | 58.55 | 46.05 | 33.29 | 27.76 | 27.52 | 27.48 | 21.37 | 21.06 | 16.04 | 31.01 | +32.05 |
| **SAFA-SNN** | **76.03** | **70.91** | **66.1** | **61.55** | **58.19** | **55.22** | **52.77** | **50.33** | **48.09** | **56.59** | |

Table 5: Comparison with FSCIL baselines on static dataset CIFAR-100 with 5way-5shot incremental learning setting and Spiking VGG-11.

| Method | Accuracy in each session (%) ↑ | | | | | | | | | $\mathcal{A}_{\text{avg}}$ ↑ | $\Delta_{\text{last}}$ |
|---|---|---|---|---|---|---|---|---|---|---|---|
| | 0 | 1 | 2 | 3 | 4 | 5 | 6 | 7 | 8 | | |
| CEC | 64.37 | 59.85 | 55.89 | 52.04 | 48.88 | 46.37 | 43.92 | 41.72 | 39.60 | 50.63 | +7.30 |
| FACT | 61.98 | 57.48 | 53.54 | 50.05 | 47.06 | 44.48 | 42.13 | 39.92 | 38.02 | 48.62 | +8.88 |
| S3C | 45.28 | 24.80 | 14.40 | 16.20 | 13.60 | 15.80 | 22.00 | 18.00 | 20.28 | +28.90 |
| SAVC | 62.95 | 58.11 | 53.96 | 50.36 | 47.21 | 44.44 | 41.97 | 39.76 | 37.77 | 48.39 | +9.13 |
| TEEN | 64.30 | 59.79 | 55.73 | 51.91 | 48.86 | 46.14 | 43.68 | 41.42 | 39.34 | 50.79 | +7.56 |
| WARP | 62.28 | 57.95 | 53.99 | 50.41 | 47.25 | 44.69 | 42.34 | 40.20 | 38.28 | 48.16 | +8.62 |
| CLOSER | 67.87 | 63.52 | 59.57 | 56.15 | 52.83 | 49.89 | 47.54 | 45.43 | 43.38 | 54.02 | +3.52 |
| **SAFA-SNN** | **76.95** | **71.20** | **66.00** | **61.81** | **58.14** | **54.74** | **51.94** | **49.16** | **46.90** | **58.12** | |

Table 6: Comparison with FSCIL baselines on static dataset CIFAR-100 with 5way-5shot incremental learning setting and Spiking ResNet-20.

| Method | Accuracy in each session (%) ↑ | | | | | | | | | $\mathcal{A}_{\text{avg}}$ ↑ | $\Delta_{\text{last}}$ |
|---|---|---|---|---|---|---|---|---|---|---|---|
| | 0 | 1 | 2 | 3 | 4 | 5 | 6 | 7 | 8 | | |
| CEC | 12.95 | 12.22 | 11.27 | 10.55 | 10.04 | 9.48 | 8.93 | 8.53 | 8.15 | 10.43 | +38.75 |
| FACT | 64.03 | 56.51 | 53.11 | 49.77 | 47.31 | 44.51 | 42.53 | 40.86 | 39.09 | 47.13 | +7.81 |
| S3C | 49.15 | 26.40 | 22.80 | 22.20 | 22.80 | 22.00 | 25.20 | 33.40 | 30.40 | 27.38 | +16.5 |
| TEEN | 65.68 | 57.79 | 54.64 | 51.07 | 48.30 | 45.53 | 43.69 | 42.00 | 40.25 | 49.21 | +6.65 |
| WARP | 46.85 | 36.85 | 34.39 | 31.56 | 29.64 | 27.88 | 26.40 | 25.25 | 23.93 | 31.64 | +22.97 |
| CLOSER | 56.02 | 53.43 | 50.33 | 47.21 | 44.65 | 42.48 | 40.44 | 38.86 | 37.10 | 45.61 | +9.80 |
| **SAFA-SNN** | **76.95** | **71.20** | **66.00** | **61.81** | **58.14** | **54.74** | **51.94** | **49.16** | **46.90** | **58.12** | |

Table 7: Comparison with FSCIL baselines on static dataset CIFAR-100 with 5way-5shot incremental learning setting and Spiking ResNet-19.

| Method | Accuracy in each session (%) ↑ | | | | | | | | | $\mathcal{A}_{\text{avg}}$ ↑ | $\Delta_{\text{last}}$ |
|---|---|---|---|---|---|---|---|---|---|---|---|
| | 0 | 1 | 2 | 3 | 4 | 5 | 6 | 7 | 8 | | |
| FACT | 67.67 | 62.17 | 58.03 | 54.41 | 51.23 | 48.64 | 46.17 | 44.14 | 42.09 | 53.28 | +4.17 |
| TEEN | 66.71 | 61.71 | 57.32 | 53.69 | 50.35 | 47.67 | 45.13 | 43.14 | 41.03 | 52.46 | +5.23 |
| WARP | 66.58 | 6.12 | 2.94 | 3.47 | 2.46 | 1.91 | 1.87 | 1.92 | 1.30 | 10.83 | +44.96 |
| **SAFA-SNN** | **73.98** | **68.74** | **63.99** | **59.99** | **56.38** | **53.49** | **51.03** | **48.60** | **46.26** | **56.35** | |

Table 8: Comparison with FSCIL baselines on static dataset CIFAR-100 with 5way-5shot incremental learning setting and Spikingformer.

| Method | Accuracy in each session (%) ↑ | | | | | | | | | $\mathcal{A}_{\text{avg}}$ ↑ | $\Delta_{\text{last}}$ |
|---|---|---|---|---|---|---|---|---|---|---|---|
| | 0 | 1 | 2 | 3 | 4 | 5 | 6 | 7 | 8 | | |
| TEEN | **68.37** | **52.42** | 48.67 | 45.43 | 42.59 | 40.08 | 37.86 | 35.86 | 34.07 | 45.37 | +2.48 |
| WARP | 63.30 | 6.45 | 4.04 | 4.20 | 2.48 | 3.34 | 2.36 | 2.35 | 2.59 | 10.38 | +33.96 |
| CLOSER | 23.53 | 21.86 | 20.24 | 18.87 | 17.95 | 16.92 | 15.97 | 15.20 | 14.47 | 18.89 | +22.08 |
| **SAFA-SNN** | 62.65 | 52.09 | **48.90** | **46.01** | **43.65** | **41.34** | **39.70** | **38.25** | **36.55** | **45.68** | |

# F  SUBSPACE PROJECTION

Prototypical networks are a few-shot learning framework designed to recognize novel classes using only a small number of labeled examples. The model uses a shared feature extractor to embed all input images into a common feature space. For each class, the embeddings of its support samples are averaged to form a prototype vector that represents the class in this space. Classification is performed by measuring the distance between query embeddings and class prototypes, typically using the Euclidean (L2) distance. Each query sample is assigned to the class whose prototype is closest in the embedding space (Wertheimer & Hariharan, 2019). As classification in prototype-based networks is non-parametric, training primarily focuses on optimizing the feature extractor. The model is trained task-wise, consisting of small N-way K-shot, enabling it to produce discriminative prototypes from limited examples. Under such a few-shot constraint, learning complex parametric decision functions is inherently difficult. In SAFA-SNN, we employ a simple yet effective prototype update strategy based on subspace projection, which isolates attribute-relevant information without introducing auxiliary networks or additional weighted loss terms (Li et al., 2020).

# G  DATASETS AND SPIKING ARCHITECTURE DETAILS

## G.1  DATASETS

The following static datasets for classification are commonly used in FSCIL.

- *CIFAR-100* (Krizhevsky et al., 2009): It contains 100 classes, with 500 training images and 100 testing images per class. Each of the 600 RGB images per class has a size of $32 \times 32$.
- *Mini-ImageNet* (Russakovsky et al., 2015): It contains 60000 RGB images of size $84 \times 84$ pixels, sampled from ImageNet-1K.

Neuromorphic datasets exhibit sparse features, typically obtained from event-based simulators or converted from frame-based datasets.

- *CIFAR10-DVS* (Li et al., 2017): A neuromorphic dataset derived from the original CIFAR10, where visual inputs are recorded by a Dynamic Vision Sensor (DVS), capturing changes in pixel intensity as asynchronous events instead of static frames. It contains 9,000 training samples and 1,000 test samples.
- *DVS128 Gesture* (Amir et al., 2017): A gesture recognition dataset consisting of 11 hand gesture categories collected from 29 individuals under three different lighting conditions.
- *N-Caltech101* (Orchard et al., 2015): A dataset comprises 8,246 event streams, each 300 milliseconds in duration, recorded by an event-based camera as it captured dynamic visual

Table 9: Comparison with FSCIL baselines on static dataset MiniImageNet with 5way-5shot incremental learning setting and Spiking VGG-5.

| Method | Accuracy in each session (%) ↑ | | | | | | | | | $\mathcal{A}_{avg}$ ↑ | $\Delta_{last}$ |
|---|---|---|---|---|---|---|---|---|---|---|---|
| | 0 | 1 | 2 | 3 | 4 | 5 | 6 | 7 | 8 | | |
| CEC | 57.28 | 52.93 | **49.98** | 47.12 | 44.90 | 42.76 | 40.29 | 38.77 | 37.57 | 45.73 | +1.06 |
| FACT | 54.51 | 43.91 | 41.16 | 39.27 | 37.68 | 35.80 | 33.95 | 32.64 | 31.82 | 38.97 | +6.81 |
| S3C | 41.01 | 12.56 | 17.43 | 29.85 | 24.01 | 25.87 | 21.32 | 30.95 | 28.50 | 25.28 | +10.13 |
| TEEN | 55.32 | 44.89 | 42.23 | 41.12 | 39.19 | 37.49 | 35.59 | 34.34 | 33.34 | 40.72 | +5.29 |
| WARP | 50.24 | 25.84 | 24.13 | 22.34 | 21.00 | 19.97 | 18.75 | 18.38 | 17.66 | 24.70 | +20.97 |
| CLOSER | 48.07 | 44.54 | 41.89 | 40.16 | 38.28 | 36.41 | 34.54 | 33.39 | 32.67 | 38.66 | +5.96 |
| **SAFA-SNN** | **61.31** | **53.06** | 49.92 | **48.12** | **46.19** | **43.88** | **41.40** | **39.86** | **38.63** | **46.93** | |

Table 10: Comparison with FSCIL baselines on static dataset MiniImageNet with 5way-5shot incremental learning setting and Spiking ResNet-18.

| Method | Accuracy in each session (%) ↑ | | | | | | | | | $\mathcal{A}_{avg}$ ↑ | $\Delta_{last}$ |
|---|---|---|---|---|---|---|---|---|---|---|---|
| | 0 | 1 | 2 | 3 | 4 | 5 | 6 | 7 | 8 | | |
| CEC | 55.23 | 51.07 | 47.62 | 45.15 | 42.84 | 40.64 | 38.42 | 36.64 | 35.03 | 43.63 | +13.06 |
| TEEN | 54.53 | 47.59 | 44.54 | 42.19 | 40.14 | 38.03 | 36.06 | 34.53 | 33.11 | 41.41 | +14.98 |
| CLOSER | 56.93 | 52.77 | 49.16 | 46.72 | 44.31 | 42.12 | 40.04 | 38.47 | 37.17 | 45.97 | +10.92 |
| **SAFA-SNN** | **76.03** | **70.91** | **66.1** | **61.55** | **58.19** | **55.22** | **52.77** | **50.33** | **48.09** | **56.59** | |

inputs of Caltech101 (Fei-Fei et al., 2004) images displayed on an LCD screen. These recordings span 101 object categories, preserving the diversity of the original dataset while incorporating temporal event-based representations.

We provide the implementation of neuromorphic dataset splitting methods for FSCIL in the code.

### G.2 SPIKING ARCHITECTURE

We transform three popular ANN architectures, i.e., VGG, ResNet, and Transformers, into their spiking neural network (SNN) equivalents, removing the need for floating-point multiplication. Specifically, we configure the Spikingformer with a depth of 2, five tokenizer convolutional blocks, and a spike-form dimensionality of 128.

We directly encode spikes with layers of LIF neurons. Each formulated unit $\mathbf{S}$ generating spikes can be represented as

$$\mathbf{S} = \mathcal{SN}((\text{BN}(\text{CONV}(\boldsymbol{X})))), \tag{42}$$

where $\boldsymbol{X} \in \mathbb{R}^{T \times B \times C \times H \times W}$ is the input with time, $\text{CONV}(\cdot)$ is the convolution layer, $\text{BN}(\cdot)$ is the batch normalization layer and $\mathcal{SN}(\cdot)$ is the LIF neuron model.

## H DETAILS OF BASELINES DESCRIPTION

We establish several baselines, including methods based on SNNs and ANNs, to better evaluate our proposed framework. We briefly list the existing SOTA methods from the FSCIL literature that we convert into SNN-based architectures for evaluation.

- *CEC* (Zhang et al., 2021): A framework with dynamically evolving architectures incrementally transforms newly added linear classifiers into a graph-based structure.

- *FACT* (Zhou et al., 2022): A framework facilitates forward compatibility by synthesizing virtual prototypes during the base training stage, thereby enabling more flexible adaptation to future tasks.

- *S3C* (Kalla & Biswas, 2022): A framework expands its stochastic classifiers by progressively incorporating four angular representations.

- *BIDIST* (Zhao et al., 2023): This method assigns a learnable weight $\mathbf{W}_t$ to each task $t$ and employs bilateral distillation between the representations of current and previous tasks, simultaneously accommodating novel concepts and retaining base knowledge.

- *SAVC* (Song et al., 2023): A framework advances base task performance through leveraging semantic-aware fantasy classes combined with contrastive learning, which enhances feature representation.

- *TEEN* (Wang et al., 2023): A prototype-tuning based approach presents a training-free prototype rectification technique, which effectively improves classification robustness without requiring additional training.

- *WARP* (Kim et al., 2023): A framework trains backbone and classifier on base tasks, then fine-tunes classifier and subnetworks of $\theta$ with multi-axis rotations for distinct representations.

- *CLOSER* (Oh et al., 2024): A framework strikes an effective balance between transferability and discriminability by employing a mechanism of compressed feature spreading, thereby improving generalization across tasks.

The following three methods are SNN-based approaches, for which we adapt the data loading strategy to the FSCIL setting for evaluation.

- *Early Bird* (Kim et al., 2022): A pruning technique for deep SNNs inspired by the Lottery Ticket Hypothesis, extracts sparse subnetworks that retain performance comparable to full dense networks.

- *SLTT* (Meng et al., 2023): An SNN training technique computes error signals independently without relying on information from other time steps, substantially reducing the required memory overhead.

- *ALADE-SNN* (Ni et al., 2025): An SNN-based framework for class-incremental learning incorporates feature extractor structure expansion and adaptive logit alignment to accommodate new tasks.

The recent advancement in FSCIL introduces parameter-efficient prompt-based methods with ViT as the backbone, demonstrating the strong capability of representation.

- *ASP* (Liu et al., 2024a): It efficiently fine-tunes small prompts on a frozen backbone, leveraging task-aware and task-invariant prompts to improve selection during testing.

- *CoACT* (Roy et al., 2024): It enables FSCIL in foundation models by employing asynchronous contrastive tuning with LoRA modules and consistency regularization to mitigate catastrophic forgetting.

- *PriViLege* (Park et al., 2024): A framework that leverages large pre-trained vision-language models for FSCIL through pre-trained knowledge tuning and specialized distillation losses.

Table 11: Performance comparison under different SEW ResNet architectures.

| Method | Architecture | $\mathcal{A}_{\text{last}} \uparrow (\%)$ | $\mathcal{A}_{\text{avg}} \uparrow (\%)$ |
|---|---|---|---|
| SEW ResNet-18 | TEEN | $38.64_{\pm 0.58}$ | $48.20_{\pm 0.74}$ |
| | WARP | $35.37_{\pm 0.37}$ | $45.43_{\pm 0.33}$ |
| | SAFA-SNN | $\mathbf{39.76}_{\pm 0.32}$ | $\mathbf{49.95}_{\pm 0.29}$ |
| SEW ResNet-34 | TEEN | $38.45_{\pm 0.28}$ | $48.14_{\pm 0.29}$ |
| | WARP | $36.35_{\pm 0.10}$ | $46.63_{\pm 0.11}$ |
| | SAFA-SNN | $\mathbf{39.12}_{\pm 0.13}$ | $\mathbf{49.46}_{\pm 0.11}$ |
| SEW ResNet-50 | TEEN | $37.35_{\pm 0.16}$ | $46.74_{\pm 0.21}$ |
| | WARP | $26.14_{\pm 1.30}$ | $34.36_{\pm 1.75}$ |
| | SAFA-SNN | $\mathbf{38.94}_{\pm 0.41}$ | $\mathbf{49.00}_{\pm 0.47}$ |

# I   MORE COMPARATIVE RESULTS

## I.1   COMPARISON OF ALL-SESSION ACCURACY

Comparisons with SOTA methods on CIFAR-100 using Spiking VGG-9, VGG-11, Spiking ResNet-20, Spiking ResNet-19, and Spikingformer are reported in Table 4, 5, 6, 7, and 8 and on Mini-ImageNet with Spiking VGG-5, Spiking ResNet-18 are listed in Table 9 and 10. We observe that as the layer increases, the performance of Spiking ResNet drops significantly, e.g., the accuracy of Spiking ResNet-34 is much lower than that of Spiking ResNet-18. The SEW residual connections support identity mapping (Fang et al., 2021). Specifically, we have conducted experiments on SEW-ResNet, as shown in Table 11. The results show that the SEW residual block, which reconstructs the residual learning and activation order using an element-wise function, prevents performance degradation as the network depth increases. These results demonstrate that SAFA-SNN consistently achieves the highest final-session performance, highlighting its robustness and effectiveness.

Table 12: Comparison with Soft SubNetwork FSCIL baselines on static dataset CIFAR-100 with 5way-5shot incremental learning setting.

| Method | Architecture | Accuracy in each session (%) ↑ | | | | | | | | | $\mathcal{A}_{avg}$ ↑ | $\Delta_{last}$ | Trainable Param |
|--------|-------------|------|------|------|------|------|------|------|------|------|------|------|------|
| | | 0 | 1 | 2 | 3 | 4 | 5 | 6 | 7 | 8 | | | |
| FSLL | ResNet-18 | 64.10 | 55.85 | 51.71 | 48.59 | 45.34 | 43.25 | 41.52 | 39.81 | 38.16 | 47.59 | +10.25 | $\mathbf{P}^t_{ST}, \mathbf{\Theta}_R$ |
| SoftNet | ResNet-18 | 72.62 | 67.31 | 63.05 | 59.39 | 56.00 | 53.23 | 51.06 | 48.83 | 46.63 | 57.57 | +1.78 | $\mathbf{m}_{soft}$ |
| WSN | ResNet-18 | 75.70 | 25.20 | 40.60 | 32.30 | 40.50 | 43.80 | 40.80 | 45.10 | 37.10 | 42.34 | +11.31 | $\theta_*, \mathbf{m}_{soft}$ |
| **SAFA-SNN** | Spiking ResNet-18 | **77.57** | **72.32** | **67.41** | **62.79** | **59.35** | **56.31** | **53.63** | **50.92** | **48.41** | **60.97** | - | - |

## I.2   COMPARISON WITH SOFT-SUBNETWORK BASED FSCIL METHODS AND SPARSITY-AWARE NEURONAL DYNAMICS

We compare the performance of SAFA-SNN with prior soft subNetwork-based FSCIL works on the CIFAR-100 dataset, including FSLL (Mazumder et al., 2021), SoftNet (Kang et al., 2023), and WSN (Kang et al., 2024) in Table 12. Specifically, in the last column, we report the additional trainable parameters required (beyond the backbone model parameters). For FSLL, the extra parameters come from the session-specific trainable parameters $\mathbf{P}^t_{ST}$ and the rotation prediction network $\mathbf{\Theta}_R$. SoftNet trains an additional soft mask $\mathbf{m}_{soft}$ to select subnetworks in the base session. WSN trains a set of parameters $\theta_*$ and $\mathbf{m}_{soft}$ that regularize the weights of the learned SubNetworks. In SAFA-SNN, the trainable $\theta$ is the intrinsic parameters of the LIF neuron model that are trained with membrane potential, and the mask $\mathbf{M}$ is used for initialization and then remains fixed. Consequently, SAFA-SNN introduces no additional trainable parameters while achieving the best performance among all compared methods.

Table 13: Performance Comparison across Sessions ($\mathcal{A}_h$ / $\mathcal{A}_n$).

| Session | 1 | 2 | 3 | 4 | 5 | 6 | 7 | 8 |
|---------|---|---|---|---|---|---|---|---|
| TEEN | 36.05 / 25.00 | 29.07 / 18.80 | 26.40 / 16.67 | 25.53 / 16.00 | 24.10 / 14.92 | 24.74 / 15.43 | 24.57 / 15.31 | 24.06 / 14.95 |
| SAFA-SNN | 41.90 / 28.80 | 34.51 / 22.30 | 29.25 / 18.13 | 27.13 / 16.55 | 27.74 / 17.04 | 27.30 / 16.73 | 26.53 / 16.17 | 27.14 / 16.68 |
| $\Delta$ | +5.85 / +3.80 | +5.44 / +3.50 | +2.85 / +1.46 | +1.60 / +0.55 | +3.64 / +2.12 | +2.56 / +1.30 | +1.96 / +0.86 | +3.08 / +1.73 |

## I.3   HARMONIC MEAN ACCURACY

Table 13 reports the harmonic mean accuracy and novel class accuracy comparison against the runner-up method on CIFAR-100 with Spiking VGG-9, further emphasizing the consistent performance improvements and the effectiveness of SAFA-SNN. These results demonstrate that base classes contain abundant semantic information, enabling well-formed prototypes for novel classes with strong prior knowledge after subspace projection and update.

## J    THEORETICAL ENERGY CONSUMPTION

According to previous studies (Horowitz, 2014; Yao et al., 2023; Lv et al., 2024), for SNNs, the theoretical energy consumption of layer $l$ can be calculated as:

$$E(l) = E_{AC} \times SOPs(l), \qquad (43)$$

where SOPs is the number of spike-based accumulate (AC) operations.

For traditional artificial neural networks (ANNs), the theoretical energy consumption required by layer $b$ can be estimated by:

$$E(b) = E_{MAC} \times FLOPs(b), \qquad (44)$$

where FLOPs denotes the number of floating-point multiply-and-accumulate (MAC) operations. We assume that both MAC and AC operations are implemented on 45nm hardware (Yao et al., 2023), where $E_{MAC} = 4.6$pJ and $E_{AC} = 0.9$pJ. Note that $1$ J $= 10^3$ mJ $= 10^{12}$ pJ.

The number of synaptic operations at the layer $l$ of an SNN is estimated as:

$$SOPs(l) = T \times \zeta \times FLOPs(l), \qquad (45)$$

where $T$ is the number of time steps in the simulation, $\zeta$ is the firing rate of the input spike trains at layer $l$.

Based on these theoretical analysis, we report the theoretical inference energy consumption results with variable structures in Table 14, which indicates that ANNs consume substantially higher energy than SNNs. Moreover, the disparity becomes more pronounced as the model size increases, which inherently limits the ability of ANNs to be implemented on edge devices.

Table 14: Inference Energy Consumption Comparison of ANNs and SNNs.

| Model | Energy (J) | Model | Energy (J) |
|---|---|---|---|
| Spiking VGG-5 | **7916.53** | VGG-5 | 10115.57 |
| Spiking VGG-9 | **3958.41** | VGG-9 | 5057.97 |
| Spiking VGG-11 | **494.92** | VGG-11 | 632.39 |
| Spiking VGG-13 | **495.00** | VGG-13 | 5689.97 |
| Spiking VGG-16 | **495.08** | VGG-16 | 5689.97 |

## K    TIME EFFICIENCY ON SAFA-SNN

To further validate the scalability and training efficiency of our method, we conducted additional experiments measuring training time on more complex model architectures. Evaluations on larger architectures, summarized in Table 15, indicate that in deeper SpikingFormer models, SAFA-SNN reduces the training time by approximately 80% compared to baseline methods, demonstrating its efficiency in complex architectures.

Table 15: Comparison of computation time across datasets and architectures.

| Dataset | Architecture | TEEN (SNN) | CLOSER (SNN) | SAFA-SNN |
|---|---|---|---|---|
| Mini-ImageNet | Spiking ResNet-18 | 104.96s | 118.24s | **95.96s** |
| CIFAR-100 | Spiking ResNet-19 | 143.17s | 372.57s | **47.80s** |
| CIFAR-100 | Spiking ResNet-20 | 93.85s | 111.06s | **69.34s** |
| CIFAR-100 | Spikingformer | 233.20s | 625.67s | **47.80s** |

## L    EFFECT ON N-WAY K-SHOT

SAFA-SNN leverages N-way K-shot datasets to estimate prototypes for novel classes. To evaluate the impact of the number of shots on accuracy, we fix the incremental way and vary shots {1,2,5,10,15,20,50} on DVS128 Gesture and CIFAR10-DVS. We can infer that with more instances per class, the estimation of prototypes will be more precise, and the performance will correspondingly improve.

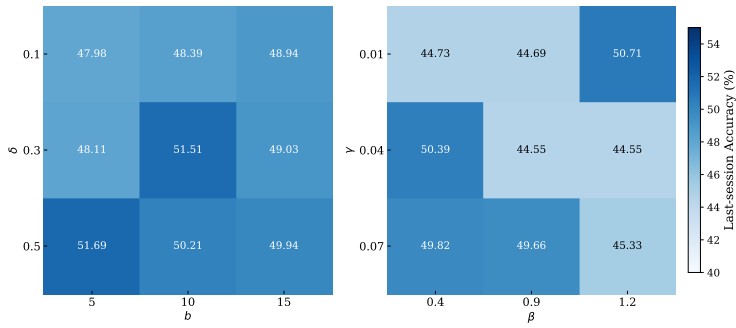

Figure 8: Comparison of different hyper-parameters on $\delta$, $b$, $\gamma$ and $\beta$ for on-device FSCIL.

## M   HYPER-PARAMETER SENSITIVITY

We report the last-session accuracy on CIFAR-100 varying the $\beta$ in $\{0.4,0.9,1.2\}$ and $\gamma$ in $\{0.01, 0.04,0.07\}$. We also change the sampled point number $b$ from $\{5,10,15\}$ and constant $\delta$ from $\{0.1,0.3,0.5\}$, resulting compared results in Figure 8. Our method is robust to the choices of hyper-parameters, and it achieves the best performance when $\delta = 0.5$ and $b = 5$, $\gamma = 0.01$ and $\beta = 1.2$. The adaptive ratio $\eta$ varies in $\{0.1, 0.3, 0.5, 0.7, 0.9\}$ and parameter $\lambda$ in $\{0.01, 0.05, 0.1\}$, as shown in Table 16. The results suggest that the performance remains stable across variations of $\lambda$ and $\eta$.

Table 16: Comparison on various $\eta$ and $\lambda$.

| $\lambda$ | $\eta$ | Accuracy in each session (%) ↑ | | | | | | | | | $\mathcal{A}_{\mathrm{avg}}$ ↑ |
|---|---|---|---|---|---|---|---|---|---|---|---|
| | | 0 | 1 | 2 | 3 | 4 | 5 | 6 | 7 | 8 | |
| 0.05 | 0.1 | 76.12 | 70.06 | 65.31 | 60.76 | 57.50 | 54.65 | 51.99 | 48.96 | 45.77 | 59.01 |
| 0.05 | 0.3 | 76.12 | 70.51 | 65.44 | 60.83 | 57.81 | 54.72 | 52.30 | 49.44 | 46.63 | 59.31 |
| 0.05 | 0.5 | **77.93** | **72.69** | **68.16** | **64.09** | **60.49** | **57.35** | **54.80** | **52.20** | **49.88** | **61.96** |
| 0.05 | 0.7 | 76.12 | 70.57 | 65.89 | 61.71 | 58.09 | 55.74 | 52.33 | 49.85 | 47.60 | 59.77 |
| 0.05 | 0.9 | 75.18 | 69.45 | 64.97 | 60.84 | 57.14 | 54.20 | 51.74 | 49.23 | 47.21 | 58.88 |
| 0.01 | 0.5 | 76.73 | 70.88 | 66.41 | 61.73 | 59.04 | 55.67 | 53.24 | 50.40 | 45.40 | 59.95 |
| 0.1 | 0.5 | 76.67 | 70.59 | 66.11 | 61.43 | 58.51 | 55.02 | 52.13 | 49.52 | 45.40 | 59.49 |

## N   EFFECTS ON DIFFERENT TIME STEPS

Upon proving the importance of SAFA-SNN modules, we further evaluate the impacts of the introduced time steps in SNN. We choose time steps $T$ among 1,2,4,6,8 and 10. We report the performance in incremental sessions on CIFAR-100 and Mini-ImageNet in Figure 9, respectively. As illustrated in the figures, the best performance is observed when $T$ is approximately between 4 and 6. Since the larger time step often corresponds to challenges like gradient vanishing or exploding (Yu et al., 2024a), we suggest $T = 4$ as the default in our experiment.

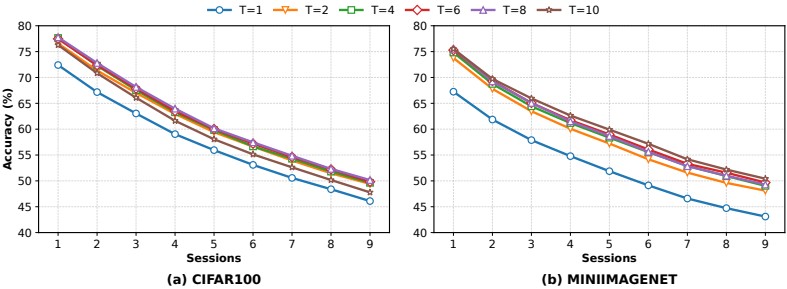

Figure 9: Accuracy in each session on different time steps.

## O    ACCURACY AND SPARSITY TRADE-OFF

We trained the CIFAR-100 dataset on the original Spiking VGG network with 285,448 neurons in all layers with dynamic threshold. We found that the training is very robust to even extreme values of adaptive ratio, as shown in Figure 10 and 11. The points with a red dot and a black border are the start point in each setting, while the points with a green dot and a black border are the end point. We can see that both the sparsity and accuracy converge at a high level, showing the promise in the sparsity-accuracy trade-off in SAFA-SNN.

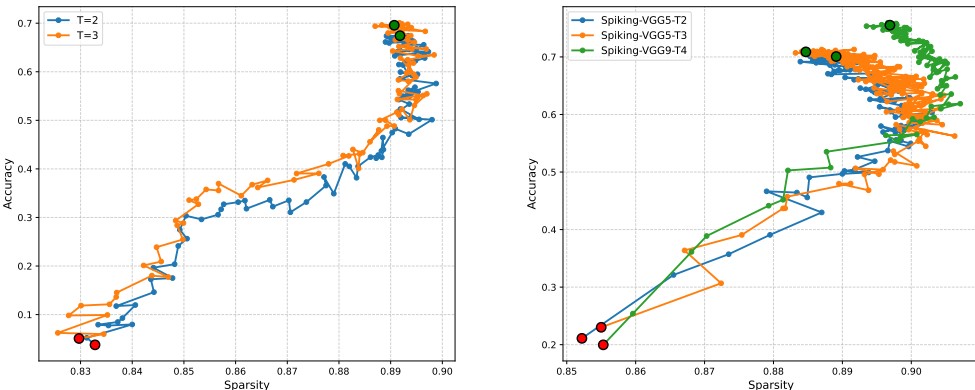

Figure 10: Sparsity-Accuracy on MiniImageNet.    Figure 11: Sparsity-Accuracy on CIFAR-100.

## P    MORE EXPERIMENTAL RESULTS

### P.1    COMPARISON WITH SURROGATE GRADIENT METHODS AND ZEROTH-ORDER OPTIMIZATION

We choose the following surrogate gradients (SG): triangle-like surrogate function (Deng et al., 2022), Piecewise-Exponential surrogate function (Neftci et al., 2019), PiecewiseLeakyReLU (Cheng et al., 2020), QPseudoSpike (Herranz-Celotti & Rouat, 2022), Sigmoid, and ATan gradient. Table 17 reports the performance differences when training SNNs with ZOO and different SG methods. Compared to the SG, ZOO estimates gradients using only function values and avoids computationally expensive high-order derivatives in SNNs (Liu et al., 2018). Due to the narrow width of SG, neuron membrane potentials often enter saturation regions where the surrogate derivative is nearly zero, causing gradient vanishing or mismatch and widening the gap from the true gradients (Lian et al., 2023). Although ZOO inevitably introduces some computational overhead from the randomly generated $b$ samples, we verify that using only five samples is adequate to maintain effectiveness in experiments. Consequently, the induced overhead is negligible. Developing a better ZOO in SNN to reduce computational overhead in the latent space would likely make SNN training easier, and we plan to explore it in future work.

Table 17: Comparison of ZOO with six surrogate gradients on CIFAR-100.

|  | Triangle | Piecewise Exp | Piecewise Leaky ReLU | QPseudoSpike | Sigmoid | ATan | ZOO |
|---|---|---|---|---|---|---|---|
| $\mathcal{A}_{\text{last}}$ (%) | $46.71_{\pm 0.32}$ | $40.98_{\pm 0.15}$ | $45.30_{\pm 0.56}$ | $41.15_{\pm 4.17}$ | $45.91_{\pm 0.19}$ | $45.59_{\pm 0.71}$ | $\mathbf{50.13}_{\pm 0.37}$ |
| $\mathcal{A}_{\text{avg}}$ (%) | $59.25_{\pm 0.12}$ | $52.76_{\pm 0.17}$ | $57.58_{\pm 0.74}$ | $54.71_{\pm 1.59}$ | $58.24_{\pm 0.29}$ | $57.92_{\pm 0.66}$ | $\mathbf{61.75}_{\pm 0.62}$ |

### P.2    COMPARISON WITH PROJECTION RELATED FSCIL METHODS AND PROTOTYPE SUBSPACE PROJECTION

Subspace projection is a training-free technique that does not incorporate any trainable parameters. Owing to the inference-only prototypes and orthogonalization-based subspace projection, SAFA-SNN can strengthen the distinctiveness of the scarce new class prototypes. The weight space rotation in WARP (Kim et al., 2023) constructs a new weight space to identify important parameters.

Subspace-reg (Akyürek et al., 2022) is a subspace regularization scheme, focusing on complicated trainable modules to pull weight vectors for new classes to lie close to the subspace spanned by the weights of existing classes. From Table 18 we can infer that SAFA-SNN can achieve not only higher performance than previous projection-based techniques for FSCIL, especially in the last incremental session, but also lead to negligible time cost, compared to previous works that require heavy training.

Table 18: Comparison of SAFA-SNN subspace projection with prior methods in terms of performance and run-time cost.

| Method | Dataset | Architecture | Execution Time | $\mathcal{A}_{\text{avg}}$ (%) | $\mathcal{A}_{\text{last}}$ (%) |
|---|---|---|---|---|---|
| Weight Space Rotation | CIFAR-100 | Spiking VGG-9 | 196.88ms | 28.74 | 38.94 |
| **Subspace Projection** | CIFAR-100 | Spiking VGG-9 | **0.002**ms | **62.19** | **50.39** |
| Subspace-reg | Mini-ImageNet | Spiking ResNet-18 | 40.5s | 53.04 | 39.00 |
| **Subspace Projection** | Mini-ImageNet | Spiking ResNet-18 | **0.002**ms | **61.75** | **50.89** |

## P.3 COMPARISON OF SAFA-SNN WITH ANN-BASED FSCIL METHODS

To further substantiate the advancement of the proposed SAFA-SNN, we compare our method with parameter-efficient ANN-based FSCIL approaches (ASP (Liu et al., 2024a), CoACT (Roy et al., 2024), PriViLege (Park et al., 2024)) via a systematic evaluation of accuracy, peak memory footprint (MB), parameter size (M), and energy consumption (mJ) on CIFAR-100, as summarized in Table 19. For a fair comparison, all methods employ the ViT backbone. ANN-based PEFT methods require nearly $10\times$ more memory, $20\times$ more parameters, and $10\times$ more energy, which severely limits their practicality on resource-constrained devices.

Table 19: Comprehensive Comparison with ANN-based PEFT methods on CIFAR-100.

| Method | Backbone | Memory | Param | Energy | $\mathcal{A}_{\text{base}}$ (%) | $\mathcal{A}_{\text{last}}$ (%) | $\mathcal{A}_{\text{avg}}$ (%) |
|---|---|---|---|---|---|---|---|
| ASP (ANN) | ViT | 9724.72 | 662.49 | 4017.06 | $\mathbf{75.53}_{\pm 0.56}$ | $11.79_{\pm 1.04}$ | $31.27_{\pm 1.45}$ |
| CoACT (ANN) | ViT | 16449.48 | 205.63 | 9226.95 | $47.25_{\pm 0.38}$ | $9.39_{\pm 2.28}$ | $15.43_{\pm 1.27}$ |
| PriViLege (ANN) | ViT | 1580.66 | 328.01 | 9230.48 | $67.39_{\pm 1.75}$ | $44.96_{\pm 0.76}$ | $54.58_{\pm 1.09}$ |
| **SAFA-SNN** | SNN ViT | **1035.30** | **11.08** | **424.34** | $73.73_{\pm 0.40}$ | $\mathbf{48.70}_{\pm 0.67}$ | $\mathbf{59.45}_{\pm 0.67}$ |

## Q ANALYSIS OF PERFORMANCE ON FEW-SHOT NOVEL-CLASS

Inspired by Wang et al. (2023), we analyze the prediction results in detail. We define "Misclassified to Base classes Ratio" (MBR) for new classes and "Misclassified to most similar New classes Ratio" (MNR) for base classes. As shown in Table 20, MBR is much higher than MNR, indicating that new classes are frequently misclassified as base classes more than base classes as new. The average accuracy on seen classes is substantially higher than that on unseen classes. Correspondingly, the "Correctly classified to Base class Number and to New class number" as CBN and CNN, otherwise, the "Mistakenly classified to Base class Number and to New class number" as MBN and MNN. We also report the accuracy on seen classes and unseen classes.

Table 20: Detailed prediction results of MBR (%) and MAR (%) between soft calibration-based method TEEN and SAFA on CIFAR-100.

| Method | CBN↑ | MBN↓ | MNN↓ | CNN↑ | MBR↓ | MAR↓ | Seen↑ | Unseen↑ |
|---|---|---|---|---|---|---|---|---|
| TEEN | 649778 | 4222 | 12108 | 5092 | 0.704 | **0.005** | 62.35 | 14.95 |
| SAFA-SNN | **650935** | **3065** | **10180** | **7020** | **0.592** | 0.006 | **73.57** | **15.43** |

The improved novel-class accuracy demonstrates that the proposed projection does not introduce additional bias; instead, it effectively reduces the tendency of predicting novel classes as base classes.

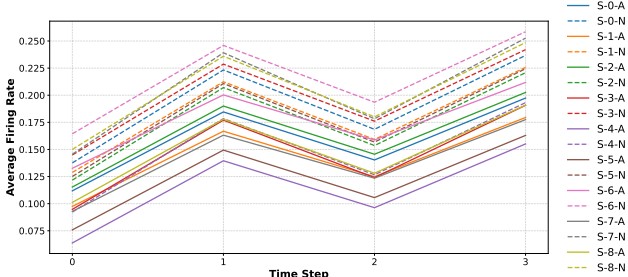

Figure 12: Firing rate of adaptive and non-adaptive neurons.

## R    VISUALIZATION OF FIRING RATE

We visualize the average firing rates in each time step of stable (dashed lines) neurons and adaptive neurons (solid lines) on CIFAR-100 dataset in Figure 12, where "S-k-A" and "S-k-N" denote the firing rate of adaptive neurons in the $k$ session and non-adaptive neurons in the $k - th$ session, respectively. It is observed that non-adaptive neurons exhibit higher firing rates than adaptive neurons in the same session. This implies that adaptive neurons constrain their activity to remain within threshold limits, potentially reducing their responsiveness to novel classes, whereas non-adaptive neurons sustain stable firing behavior to mitigate the risk of catastrophic forgetting.

## S    DISCUSSION

Our method significantly improves the recognition accuracy of new classes in on-device FSCIL scenarios, where catastrophic forgetting and resource constraints severely limit model performance. By addressing the underperformance of new classes, our approach offers new insights into the overlooked challenges of class imbalance and forgetting. To the best of our knowledge, this is the first study in the on-device FSCIL literature to systematically analyze the performance degradation of new classes, highlighting the need to prioritize their evaluation in future on-device FSCIL research. In addition, our method may also offer potential heuristic effects on initiating training with sparse models for implementing edge intelligence on neuromorphic hardware.

## T    THE USE OF LLM

We only use LLM to polish some descriptive sentences and check grammatical errors.

