# OpenReview forum: "SAFA-SNN: Sparsity-Aware On-Device Few-Shot Class-Incremental Learning with Fast-Adaptive Structure of Spiking Neural Network"
_ICLR.cc/2026/Conference — ICLR 2026 Poster_

### Official Review · Reviewer_hwBr · 2025-10-28

**Soundness:** 3
**Presentation:** 3
**Contribution:** 3
**Rating:** 6
**Confidence:** 4

**Summary:**

This study focuses on the Few-Shot Class-Incremental Learning (FSCIL) problem, aiming to enable edge devices to continuously learn novel classes under realistic constraints of limited data and energy resources. To achieve this, the authors propose the SAFA-SNN framework, which comprises three main components: Sparsity-Aware Neuron Dynamics, Zeroth-Order Optimization, and Subspace Projection for prototype alignment. Notably, all experiments are implemented and rigorously evaluated on real edge devices.

**Strengths:**

1. The computation energy measurement methodology employed in this study is notably fair and well-justified.

2. This work pioneers the exploration of on-device SNN-based FSCIL, establishing a solid empirical baseline for future research.

**Weaknesses:**

1. Some equations lack proper punctuation (e.g., Eqs. 14 and 15), and the third section of Table 3 could be better centered for readability.

2. Although the Zeroth-Order Optimization (ZOO) section provides an upper bound on convergence, it lacks quantitative comparison with surrogate backpropagation and detailed analysis of potential error sources.

3. The paper mentions the degradation issue in deeper networks (e.g., ResNet-34) but does not provide a systematic explanation or proposed solution.

**Questions:**

Overall, the paper is comprehensive and well-structured; however, I would appreciate clarification on a few points:

1. Does the subspace projection introduce prototype bias toward base classes, potentially impairing performance on novel classes?

2. Is the division between active and stable neurons fixed or adaptively updated during training?

3. Why is the initial mask retained during incremental learning phases? Could this lead to over-saturation of active neurons over time?

---

> ### Author Response · Authors · 2025-11-20
> **Response to Reviewer hwBr (1/3)**
>
> Dear Reviewer hwBr,
>
> We sincerely appreciate your thoughtful comments. We have carefully considered each of your questions and provide detailed responses below.
>
> **[W1]  Some equations lack proper punctuation (e.g., Eqs. 14 and 15), and the third section of Table 3 could be better centered for readability.**
>
> **Response.** Thanks for the suggestions. Eq. 14 constructs the orthogonal projection matrix of the base prototypes, and Eq. 15 introduces a convex combination between the prototypes and their projected prototypes of novel classes. By convention, they are presented without punctuation. We have centered the third section of Table 3 in the revised manuscript to improve readability.
>
> ---
>
> **[W2]  Although the Zeroth-Order Optimization (ZOO) section provides an upper bound on convergence, it lacks quantitative comparison with surrogate backpropagation and detailed analysis of potential error sources.**
>
> **Response.** Thanks for pointing out these important problems.
>
> * **Comparison with surrogate backpropagation and ZOO**. Please refer to our response to Reviewer qRkQ's [Q2].
>
> - **Analysis of potential error sources.** We first develop the gradient approximation errors for ZOO, denoted by $‖g(x) − \nabla \Phi(x)‖$. According to [1], there are **two sources of error**: (i) approximation of the true gradient function $\Phi$ by the Gaussian smoothed function $F$ of the noisy function $f$, and (ii) approximation of $\nabla F (x)$ via sample average approximations. One common approach to optimizing functions without derivatives is to compute an estimate of the gradient $\nabla \Phi(x)$ at the point $x$, denoted by $g(x)$, using noisy function values and then apply a gradient-based method with $g(x)$. Hence, we have that
>
>   $$
>   \parallel g(x) − \nabla\Phi(x) \parallel = \parallel  (\nabla F (x) − \nabla\Phi(x)) + (g(x) − \nabla F (x))\parallel  ≤ \parallel \nabla F (x) − \nabla \Phi(x) \parallel + \parallel g(x) − \nabla F (x) \parallel
>   $$
>
>   *The bound on the first term $\parallel \nabla F (x) − \nabla \Phi(x)\parallel$.* Assume that the noise in the function evaluations $\epsilon(x)$ is bounded for all $x \in 	\mathcal{R^n}$, and that $\Phi$ is Lipschitz smooth. Assume f is an approximation of $\Phi$ with the approximation error bounded by $f$ uniformly, i.e., there is a constant $\epsilon_f ≥ 0$ such that $|f (x) − φ(x)| = | \epsilon(x)| ≤ \epsilon_f$ for all $x \in R^n$. Then the following bounds hold for the error between $\nabla F (x)$ and $\nabla \phi(x)$. If $\phi$ has L-Lipschitz continuous gradients, then
>
>   $$
>   \parallel ∇F (x) − \nabla \Phi(x) \parallel \leq \sqrt{n}L\sigma +  \frac{ \sqrt{n}\epsilon_f}{\sigma}
>   $$
>
>   If the function $\phi$ has M-Lipschitz continuous Hessians, then
>
>   $$
>   \parallel \nabla F (x) − \nabla \Phi(x) \parallel \leq nM \sigma^2 +  \frac{ \sqrt{n}\epsilon_f}{\sigma}
>   $$
>
>   *The bound of the second term $\parallel g(x) − \nabla F (x) \parallel$.* The error is due to the sample average approximation. Under the assumption of "Lipschitz continuity of the gradients of $\phi$" and "Lipschitz continuity of the Hessian of $\phi$", we have $\nabla F(x)$ is L-Lipschitz continuous and $\nabla F(x)$ is M-Lipschitz continuous. The variance of **Equation 9** in our paper can be defined as
>
>   $$
>   \operatorname{Var}\{g(x)\}
>   = \frac{1}{N} \, \mathbb{E}_{u \sim \mathcal{N}(0,I)}
>   \left[ \left( \frac{f(x+\sigma u) - f(x-\sigma u)}{2\sigma} \right)^{2} uu^{T} \right]- \frac{1}{N} \ \nabla F(x)\nabla F(x)^{T}
>   $$
>
>   Using the conclusion from [1] that when the $g(x)$ is calculated by Equation 9 in our paper, then, for all $x \in \mathbb{R}$, $Var{g(x)} \leq \kappa(x) I$ where
>
>   $$
>   \kappa(x) = \frac{3}{N} \left( 3\parallel \nabla \varphi(x) \parallel ^2 + \frac{M^2 \sigma^4}{36} (n+2)(n+4)(n+6)+ \frac{\epsilon_f^2}{\sigma^2} \right)
>   $$
>
>   In summary, with bounds for both terms, we can bound $\parallel g(x) − \nabla \Phi(x) \parallel$, in probability using Chebyshev’s inequality.

---

> ### Author Response · Authors · 2025-11-20
> **Response to Reviewer hwBr (2/3)**
>
> **[W3]  The paper mentions the degradation issue in deeper networks (e.g., ResNet-34) but does not provide a systematic explanation or proposed solution.**
>
> **Response.** We appreciate the reviewer’s constructive comment. This is based on the observation that SAFA-SNN consistently achieves SOTA results on both small-scale (Spiking VGG, Spiking ResNet-20) and complex architectures (Spikingformer, QKformer, SpikingResformer), but the advantage slightly degrades in Spiking  ResNet-34. There are two main reasons. First, in LIFspike neuron model, the sharp feature with a larger value and the non-sharp feature with a smaller value produce the same output in the forward process if the corresponding membrane potentials both exceed the firing threshold. Consequently, the information loss inherent in discrete spikes will make residual structures inapplicable to perform identity mapping. Second, the spiking transformer backbone incorporates the residual connection and self-attention mechanism in a spike-driven way, reducing the gap between them, preventing the degradation. Therefore, a promising next step is to explore the integration of self-attention with linear complexity or information augmentation between the binary model and full-precision model, potentially as a direction for developing compact yet high-performance SNNs for on-device FSCIL.
>
> ------
>
> **[Q1]  Does the subspace projection introduce prototype bias toward base classes, potentially impairing performance on novel classes?**
>
> **Response.** Thank you for this insightful question. Below, we clarify how the prototype subspace projection of our method addresses it.
>
> Prototypes learned from narrow-size distribution of scarce data of novel classes usually tend to get biased in FSCIL scenarios [3]. We figure out two key factors on the observation: (i) the average accuracy across base classes is much higher than novel classes, and novel classes tend to be misclassified as base classes (Table 5 in Appendix); (ii) the base classes contain abundant semantic knowledge, which can get well-established prototypes with adequate prior knowledge. Thus, we propose prototype subspace projection, which involves two steps to generating prototypes to update novel class classifier: (i) projecting prototypes of novel classes onto orientation subspace of base classes (Eq. 14) and (ii) blending projection prototypes $\tilde{P}_{proj}$ and novel class prototypes $\tilde{C}$ via convex combination (Eq. 15). Orthogonal projection serves as a metric to calculate the distance between the novel and base classes, where larger value of indicates a stronger cosine-similarity between them. That is, features with larger similarity hold larger proportions in prototype updating. Compared with the basic prototype $\tilde{C}$, the updated prototype distributes closer to the expected prototype and learns discriminative features more effectively. In addition, the trade-off weight factor $\alpha$ in Eq. 15  can balance the knowledge integration of base and novel classes. The improved accuracy across novel class and harmonic mean (Table 4, 5 in Appendix) further verifies the projection will not introduce bias but help diminish the bias toward base classes.
>
> ------
>
> **[Q2]  Is the division between active and stable neurons fixed or adaptively updated during training?**
>
> **Response.** The division of active and stable neurons is fixed during training. The reasons can be summarized below:
>
> - **Neuron-Class mapping.** By fixing the partition, we maintain a subset of most neurons that provide stable representations for base classes to preserve knowledge, while adaptive neurons can flexibly adjust to learn novel-class inputs.
> - **Dynamic threshold.** While the division is fixed, the thresholds of neurons are updated according to their recent behaviors (Eq. 7), and adaptive neurons update more strongly than stable ones. This allows the network to dynamically adjust firing behavior in response to novel inputs without changing the underlying division.
>
> ------
>
> **[Q3]  Why is the initial mask retained during incremental learning phases? Could this lead to over-saturation of active neurons over time?**
>
> **Response.** Thanks for pointing out these concerns. The initial mask is used only to randomly initialize the neuron partition. Considering the dynamic nature of FSCIL, where new classes arrive sequentially and the mask needs to be updated in each incremental session, making the mask trainable would introduce additional computational overhead and remove the training-free advantage of SAFA-SNN during incremental updates. The retained mask **will not** lead to over-saturation, because the sparsity-aware neuronal dynamics incorporate a dynamic threshold mechanism that continuously regulates the firing thresholds of overly active and overly inhibited neurons. As a result, neuron behaviors can adapt over time. For more details about this mechanism, please refer to our response to Reviewer qRkQ’s Q1.

---

> ### Author Response · Authors · 2025-11-20
> **Response to Reviewer hwBr (3/3)**
>
> **References**:
>
> [1] Berahas et al. A theoretical and empirical comparison of gradient approximations in derivative-free optimization. Foundations of Computational Mathematics 2022.
>
> [2] Nesterov et al. Random gradient-free minimization of convex functions. Foundations of Computational Mathematics 2017.
>
> [3] Liu et al. Prototype rectification for few-shot learning. ECCV 2020.

---

> ### Author Response · Authors · 2025-11-20
> **We are looking forward to your feedback.**
>
> Dear reviewer hwBr,
>
> We sincerely hope that the additional experiments and clarifications effectively address your concerns. If you have any further questions/concerns, please do not hesitate to let us know. We would greatly appreciate it if you would consider updating your ratings, particularly regarding **rating** and **technical quality** in light of our improvements.
>
> Thank you very much,
>
> Authors

---

> > ### Comment · Reviewer_hwBr · 2025-11-24
> > **It seems that more issues have emerged.**
> >
> > 1. I have reviewed the latest version of the manuscript, and it appears that the authors have not made any revisions. I still cannot understand why the authors have added punctuation marks at the end of some equations but not others. The authors seem to believe this is correct. Could the authors clarify what they mean by "convention" as mentioned in the first point of their rebuttal?
> >
> > 2. Regarding the authors' response in the second point of the rebuttal concerning identity mapping and residual structures, I believe there is a misunderstanding. Fang et al. have already demonstrated in their work on SEW-ResNet that SEW-style residual connections can achieve identity mapping and can be successfully trained to over 300 layers without degradation. However, the authors appear to be either unaware of this or consider their proof to be incorrect. Please clarify this issue.

---

> ### Author Response · Authors · 2025-11-25
> **Response to Reviewer hwBr**
>
> **Response 1.** Thank you for engaging with our response and raising these further concerns. We sincerely apologize for the confusion caused by the punctuation in the equations.
>
> In our previous reply, our reference to “convention” was unclear, and we appreciate the opportunity to clarify this point. After carefully reviewing your comments, we realized that the original submission indeed lacked consistent punctuation across equations. We have therefore updated the manuscript by applying correct and uniform punctuation throughout.
>
> A fully revised version reflecting these changes has been uploaded for your convenience. Thank you again for your careful and constructive review, which has helped improve the clarity and rigor of our work.
>
> ------
>
> **Response 2.** Thank you for your valuable feedback. We sincerely appreciate your clarification regarding SEW ResNet and fully agree with the finding by Fang et al., who demonstrated that SEW-style residual connections can achieve identity mapping and enable the training to very deep networks.
>
> We apologize for any misunderstanding caused by our previous response. Our intention was not to question the theoretical capability of SEW connections, but rather to explain the performance degradation observed in our specific implementation. The Spiking ResNet-34 model we used was directly converted from its ANN counterpart, and the structure of its residual blocks may have limited the ability of spiking neurons to realize identity mapping, leading to degradation.
>
> In light of your comments, we have revised the manuscript to explicitly acknowledge the result of Fang et al. and to clarify that SEW-style residual connections indeed support identity mapping. To further address this issue, we are conducting additional experiments with deeper SEW ResNet architectures. Due to the time required to run these experiments with multiple seeds, the results will be presented once available and included in the revised paper.
>
> | Method       | Architecture  | $ACC_{last}$   | $ACC_{avg}$    |
> | ------------ | ------------- | -------------- | -------------- |
> | TEEN         | SEW ResNet-18 | 38.64±0.58     | 48.20±0.74     |
> | WARP         | SEW ResNet-18 | 35.37±0.37     | 45.43±0.33     |
> | **SAFA-SNN** | SEW ResNet-18 | **39.76±0.32** | **49.95±0.29** |
> | TEEN         | SEW ResNet-34 | 38.45±0.28     | 48.14±0.29     |
> | WARP         | SEW ResNet-34 | 36.35±0.10     | 46.63±0.11     |
> | **SAFA-SNN** | SEW ResNet-34 | **39.12±0.13** | **49.46±0.11** |
> | TEEN         | SEW ResNet-50 | 37.35±0.16     | 46.74±0.21     |
> | WARP         | SEW ResNet-50 | 26.14±1.30     | 34.36±1.75     |
> | **SAFA-SNN** | SEW ResNet-50 | **38.94±0.41** | **49.00±0.47** |
>
> We have conducted experiments on SEW-ResNet using three different random seeds. The results show that the SEW residual block, which reconstructs the residual learning and activation order using element-wise function prevents performance degradation as the network depth increases.
>
>
> We are grateful for your careful reading and constructive feedback, which has helped us improve the clarity of our work.
>
> -----
>
>
> **References**:
>
> Fang et al. Deep residual learning in spiking neural networks. NIPS 2021.

---

### Official Review · Reviewer_smss · 2025-10-28

**Soundness:** 2
**Presentation:** 2
**Contribution:** 2
**Rating:** 4
**Confidence:** 4

**Summary:**

This paper applies spiking neural networks (SNNs) to the few-shot class-incremental learning (FSCIL) scenario and proposes SAFT-SNN, which integrates sparsity-aware neuronal dynamics, zeroth-order optimization, and fast adaptive prototype subspace projection. The author deployed the SNN on a mobile platform and conducted experiments on static and neuromorphic datasets. The results demonstrated that their method outperformed other comparative methods.

**Strengths:**

The experimental results comparing the proposed method with other approaches demonstrate its superior performance and efficiency.

**Weaknesses:**

1. Why do sparsity-aware neural dynamics allow most neurons to remain stable while activating only a few? The authors should explain the method they propose in more detail.
2. Although the author deployed the proposed method on a mobile platform, they evaluated theoretical power consumption rather than actual runtime power consumption. Moreover, the results in Figure 5 indicate that its speed does not significantly outperform other methods. This casts doubt on the true efficiency of the proposed method.
3. The proposed SAFA-SNN is nearly impossible to deploy on event-driven neuromorphic chips because Eq. (14) involves computationally intensive matrix multiplication. Therefore, it appears that the efficiency and power consumption advantages of SNNs cannot be realized.

**Questions:**

See weakness.

---

> ### Author Response · Authors · 2025-11-19
> **Response to Reviewer smss (1/2)**
>
> Dear Reviewer smss,
>
> We sincerely appreciate your thoughtful comments. We have carefully considered each of your questions and provide detailed responses below.
>
> **[W1]  Why do sparsity-aware neural dynamics allow most neurons to remain stable while activating only a few? The authors should explain the method they propose in more detail.**
>
> **Response.**  Thank you for your thoughtful comments. Neurons are associated with classes in FSCIL through a structured mapping. At the microscopic level, neurons exhibit distinct responses when learning different classes, such as activating for certain classes while being suppressed for others. This behavior accumulates to shape the final class prediction, demonstrating class-specific patterns. Based on this insight, the abundant base classes are mapped to the majority of neurons, whereas a small subset of neurons is reserved for novel classes. The dynamic threshold mechanism regulates neuronal activity: neurons representing base classes maintain stable spiking during incremental learning, aligning closely with their activity in the base class training session (session 0) to preserve prior acquired knowledge and mitigate catastrophic forgetting. Neurons associated with novel classes exhibit adaptive learning, with thresholds adjusted according to current spiking activity. Overly active neurons are inhibited through increased thresholds, while overly inhibited neurons are facilitated by lowered thresholds. Notably, the threshold adjustments for novel-class neurons are larger than those for base-class neurons, reflecting the distinct functional roles of these two neuronal populations. We have summarized sparsity-aware neuronal dynamics by formulations, please refer to our response to reviewer qRkQ's [Q4].
>
> ------
>
> **[W2]  Although the author deployed the proposed method on a mobile platform, they evaluated theoretical power consumption rather than actual runtime power consumption. Moreover, the results in Figure 5 indicate that its speed does not significantly outperform other methods. This casts doubt on the true efficiency of the proposed method.**
>
> **Response.** Thank you for the insightful comment. We would like to emphasize that our evaluation does include actual runtime power measurements, **not just** theoretical power consumption. In our original submission, we described two types of energy consumption in the “Energy Consumption” paragraph of Section 6.2: the actual energy measured during training (with results shown in Figure 4), and the theoretical energy consumption for inference (with the calculation procedure and results provided in Appendix A.9). Specifically, Figure 4 reports the actual runtime power consumption during training process, obtained through direct measurements on the Jetson Orin AGX device, which is gauged by multiplying power (W) by actual running time using the built-in sensor values. The theoretical power consumption represents the energy usage during inference, estimated from the architecture’s FLOPs, and primarily reflects the algorithmic complexity. By incorporating both theoretical and actual runtime energy measurements, we provide a more comprehensive and accurate evaluation.
>
> Thank you for raising this important question about training speed. Actually, in Figure 5, the training time is measured over the same 300 iterations and averaged for both the baselines and our SAFA-SNN to ensure a fair comparison on variant VGG structures. To further verify the scalability and training efficiency of our method, we conduct additional experiments measuring training time on more complex model architectures. These results show that our method achieved competitive accuracy and relatively shorter training time within this budget. To demonstrate this, we have included evaluations on the larger architectures, as shown in the table below. As shown in the last line of the table for deeper SpikingFormer models on the CIFAR-100 dataset, SAFA-SNN significantly reduces the training time by approximately 80% compared to baselines, demonstrating its time efficiency in complex architectures.
>
> | Dataset       | Architecture      | TEEN (SNN) | CLOSER (SNN) | **SAFA-SNN** |
> | ------------- | ----------------- | ---------- | ------------ | ------------ |
> | Mini-ImageNet | Spiking ResNet-18 | 104.96s    | 118.24s      | **95.96s**   |
> | CIFAR-100     | Spiking ResNet-19 | 143.17s    | 372.57s      | **47.80s**   |
> | CIFAR-100     | Spiking ResNet-20 | 93.85s     | 111.06s      | **69.34s**   |
> | CIFAR-100     | Spikingformer     | 233.20s    | 625.67s    | **47.80s**   |

---

> ### Author Response · Authors · 2025-11-19
> **Response to Reviewer smss (2/2)**
>
> **[W3]  The proposed SAFA-SNN is nearly impossible to deploy on event-driven neuromorphic chips because Eq. (14) involves computationally intensive matrix multiplication. Therefore, it appears that the efficiency and power consumption advantages of SNNs cannot be realized.**
>
> **Response.** We thank the reviewer for raising concerns about the computational overhead of the prototype subspace projection process. We agree that the prototype-shift module contains matrix multiplications, which are computationally more expensive than simple local SNN operations. However, we would like to clarify that this component **does not undermine the efficiency advantages of SNNs**. Below, we summarize our analysis, which demonstrates that the approach is computationally efficient and scales well for practical implementations.
>
> - We acknowledge SAFA-SNN cannot yet be fully deployed on modern neuromorphic chips. However, it can leverage a small auxiliary CPU or MCU to handle Eq.14, thereby maximizing the energy-efficiency advantages of SNNs within a heterogeneous architecture. In the future, with alternative solutions or next-generation neuromorphic hardware, it will become feasible to run small, lightweight models entirely on neuromorphic chips.
> - The prototype update is performed **only once in each incremental session for inference**, rather than at every batch or epoch, making the computational overhead highly controllable. In FSCIL, the feature extractor is frozen during incremental learning, while only the classifier can be updated. Compared to retraining the entire linear classifier, which requires repeated optimization over all samples, with cost proportional to $N × B × D × \mathcal{T}$, where $\mathcal{T}$ represents the iteration number, our subspace projection-based update is substantially more efficient. The cost of Eq. (14) depends only on the number of base classes $B$, few new classes $C$, and feature dimension $D$, dominated by the $D^3$, and is performed only once per session. Since $N>>D>B$, the one-time projection contributes a negligible portion to the overall computational cost and does not become a performance bottleneck.
> - From time efficiency aspect, the prototype subspace projection only takes negligible **0.002 ms** on average across each incremental session in FSCIL. Please refer to our response to Reviewer pCz7's [W1] for details.
>
> ------
>
> If you have any further questions/concerns, please do not hesitate to let us know. We would greatly appreciate it if you would consider updating your ratings, particularly regarding **rating** and **technical quality** in light of our improvements.
>
> Thank you very much,
>
> Authors

---

### Official Review · Reviewer_mLap · 2025-10-30

**Soundness:** 1
**Presentation:** 2
**Contribution:** 1
**Rating:** 2
**Confidence:** 5

**Summary:**

This paper proposes SAFA-SNN (Sparsity-Aware Feature Alignment), a novel framework designed to improve the efficiency and performance of Spiking Neural Networks (SNNs). The core idea is to leverage sparsity-aware alignment between membrane potentials and firing spikes, thereby improving representation stability under sparse activations. The method integrates feature alignment loss with adaptive threshold tuning to balance accuracy and energy efficiency. Extensive experiments on datasets such as CIFAR10-DVS, DVS-Gesture, and ImageNet-DVS are provided.

**Strengths:**

The paper effectively identifies a key issue in SNN training — misalignment between spike-based representations and underlying continuous features due to sparse and discrete firing dynamics. The motivation is clearly articulated and well supported by empirical visualization. The introduction of a sparsity-aware term that aligns latent representations between spike and potential spaces is interesting. It extends prior surrogate gradient ideas by explicitly modeling alignment errors as optimization objectives.

**Weaknesses:**

1. Although the “sparsity-aware alignment” terminology is new, conceptually it overlaps with prior ideas such as temporal consistency regularization and membrane potential loss. The authors should clarify in what way SAFA differs fundamentally rather than being another variant of existing potential-based regularization.
2. The sparsity-aware loss is introduced heuristically without formal derivation. It would be beneficial to provide an analysis connecting the proposed alignment loss to information preservation or stability bounds under sparse activations.
3. The ablation in Table 5 (p. 8) only evaluates the removal of individual terms, but does not show sensitivity to sparsity thresholds or alignment weights. Additionally, no visual explanation of the “feature alignment” effect is provided beyond one layer.
4. Since the method claims to be “efficiency-oriented,” a more detailed discussion on energy or FLOPs reduction would strengthen the claim. Currently, results are purely accuracy-based, with no measured power or latency comparison on neuromorphic hardware.
5. It remains unclear whether SAFA-SNN generalizes well under noise, temporal jitter, or unseen sensor dynamics. These aspects are critical for practical deployment of SNNs in event-based vision systems.
6. The paper describes the feature alignment mechanism intuitively but does not provide a clear mechanistic link between sparsity-aware alignment and improved gradient flow or spike stability. It remains unclear why aligning spike and potential features under sparsity constraints leads to consistent accuracy gains. A more detailed theoretical or empirical explanation (e.g., gradient norm analysis) would be valuable.
7. Given that the proposed loss introduces additional terms involving sparsity and alignment, it is important to analyze how this affects training convergence and gradient stability. However, there is no discussion or empirical observation (e.g., training curves or variance analysis) to ensure that the added loss does not destabilize optimization.

**Questions:**

1. How is the “sparsity-aware weighting” computed dynamically — is it global, per-layer, or per-neuron?
2. What is the relationship between SAFA and temporal consistency loss?
3. Does SAFA introduce any inference-time overhead? If not, please clarify whether alignment is training-only.
4. Can SAFA be combined with spike transformer architectures such as QKFormer or SpikingResformer?
5. Have the authors evaluated the stability of training curves (loss oscillation, firing rate variance)?

---

> ### Author Response · Authors · 2025-11-21
> **Response to Reviewer mLap (1/2)**
>
> Dear Reviewer mLap,
>
> We sincerely appreciate your thoughtful comments. We have carefully considered each of your questions and provide detailed responses below.
>
> **[W1]  Although the “sparsity-aware alignment” terminology is new, conceptually it overlaps with prior ideas such as temporal consistency regularization and membrane potential loss. The authors should clarify in what way SAFA differs fundamentally rather than being another variant of existing potential-based regularization.**
>
> **Response.** Thanks for the comments. We respectfully clarify that the term “sparsity-aware alignment” is neither used nor proposed anywhere in our paper. Our work proposes sparsity-aware neuronal dynamics for FSCIL, based on the dynamics of LIFSpike neurons’ behavior to preserve base-class knowledge, which is fundamentally different from the temporal consistency regularization and membrane potential loss.
>
> - Different from the "temporal consistency regularization". In our method, membrane potentials are updated adaptively according to neurons' previous behaviors, need not to be trained of membrane potential loss. Moreover, the temporal consistency is a key term in video data processing, is not any conterpart of the neuron dynamics in temproal locally or SNNs.
> - The proposed sparsity-aware dynamics contain two types of neurons. (i) most stable neurons: they are used to preserve the learned knowledge of base classes with relative stable firing rates (denoted by sparsity in our paper), and (ii) few adaptive neurons: they are used to learn new knowledge for FSCIL to alleviate catastrophic forgetting with relative active firing rate. Both the activities of stale neurons and adaptive neurons are controlled by the dynamic threshold. Please refer to our response to Reviewer pCz7's W2 for the comparison analysis of similar methods with our sparsity-aware neuronal dynamics.
>
> ------
>
> **[W2]  The sparsity-aware loss is introduced heuristically without formal derivation. It would be beneficial to provide an analysis connecting the proposed alignment loss to information preservation or stability bounds under sparse activations.**
>
> **Response.** We thank the reviewer for raising this point. However, we respectfully emphasize that the manuscript **does not contain any sparsity-aware loss or alignment loss**. These concepts **are not part of our method**. We would like to clarify that the loss function described in Eq. (11) is a combination of CE and MSE losses between the predictions and labels, without incorporating any additional terms. To avoid any ambiguity and to ensure that all concerns are properly addressed, we would appreciate it if the reviewer could clarify clearer which section or equation is being interpreted as a sparsity-aware loss or alignment loss.
>
> ------
>
> **[W3]  The ablation in Table 5 (p. 8) only evaluates the removal of individual terms, but does not show sensitivity to sparsity thresholds or alignment weights. Additionally, no visual explanation of the “feature alignment” effect is provided beyond one layer.**
>
> **Response.** We respectfully note that the reviewer’s statement regarding Table 5 appears to be based on a misunderstanding. Feature alignment is not part of our work, as it is beyond the scope of the paper. Furthermore, the ablation study on different combinations of our three components is reported in **Figure 6**, not Table 5 (p. 8). The terms mentioned by the reviewer are unclear and do not appear to be relevant to our method.
>
> ------
>
> **[W4]  Since the method claims to be “efficiency-oriented,” a more detailed discussion on energy or FLOPs reduction would strengthen the claim. Currently, results are purely accuracy-based, with no measured power or latency comparison on neuromorphic hardware.**
>
> We respectfully disagree with the reviewer’s statement that our method is “efficiency-oriented” and lacks power or latency comparisons. Our paper does not claim that the method is efficiency-oriented. In fact, we have conducted extensive comparative experiments with baselines across multiple metrics, including training time (Figure 5), actual energy consumption (Figure 4), theoretical energy consumption (Appendix Table 13), and sparsity-accuracy trade-offs (Appendix Figures 10 and 11). Thus, our evaluation goes beyond purely accuracy-based comparisons.
>
> ------
>
> **[W5]  It remains unclear whether SAFA-SNN generalizes well under noise, temporal jitter, or unseen sensor dynamics. These aspects are critical for practical deployment of SNNs in event-based vision systems.**
>
> **Response.** We thank the reviewer for raising this point. However, we are not aware of any prior work or publicly available datasets that define or study “unseen sensor dynamics,” and it is unclear what is specifically meant by this term. Regarding “temporal jitter,” this concept is beyond the scope of our current study.

---

> ### Author Response · Authors · 2025-11-21
> **Response to Reviewer mLap (2/2)**
>
> **[W6]  The paper describes the feature alignment mechanism intuitively but does not provide a clear mechanistic link between sparsity-aware alignment and improved gradient flow or spike stability. It remains unclear why aligning spike and potential features under sparsity constraints leads to consistent accuracy gains. A more detailed theoretical or empirical explanation (e.g., gradient norm analysis) would be valuable.**
>
> **Response.** Thank you for the comment. We would like to clarify that our paper does not introduce “sparsity-aware alignment,” “improved gradient flow,” or “spike stability” as part of the method, and it is unclear which sections the reviewer is referring to. Our work proposes **sparsity-aware neuronal dynamics** for FSCIL, which preserves base-class knowledge by controlling local firing rates, and a **novel prototype subspace projection** technique for learning new classes, addressing overfitting and bias. These components are distinct from the concepts mentioned by the reviewer.
>
> ------
>
> **[W7]  Given that the proposed loss introduces additional terms involving sparsity and alignment, it is important to analyze how this affects training convergence and gradient stability. However, there is no discussion or empirical observation (e.g., training curves or variance analysis) to ensure that the added loss does not destabilize optimization.**
>
> **Response.** Thanks for the suggestions. We will add training curves and provide analysis in our paper.
>
> ------
>
> **[Q1]  How is the “sparsity-aware weighting” computed dynamically — is it global, per-layer, or per-neuron?**
>
> **Response.**  Thanks for the question. The sparsity-aware neuronal dynamics is one of the contributions of SAFA-SNN, and it is channel-wise. But the "sparsity-aware weight" does not come from our paper.
>
> ------
>
> **[Q2]  What is the relationship between SAFA and temporal consistency loss?**
>
> **Response.**  We thank the reviewer for the question. However, our paper does not include or discuss any “temporal consistency loss,” and it is unclear what is meant by this term. There is no relation between such a loss and our proposed SAFA-SNN.
>
> ------
>
> **[Q3]  Does SAFA introduce any inference-time overhead? If not, please clarify whether alignment is training-only.**
>
> **Response.** Thanks for the good question. Since SAFA adjusts the thresholds in the inference, it may introduce negligible inference-time overhead (<1s). Once again, we would like to emphasize that our paper does not focus on any alignment.
>
> ------
>
> **[Q4]  Can SAFA be combined with spike transformer architectures such as QKFormer or SpikingResformer?**
>
> **Response.** Thanks for the insightful question. Actually, in our original submission, we have reported the comparative results on Spikingformer architecture in our original paper (See Appendix A.8 and Table 10). To show the scalability of SAFA-SNN, we report the performance results (accuracy across 9 tasks) using QKFormer [1] and SpikingResformer [2] on CIFAR-100 with the 5-way 5-shot setting in the table below.
>
>  | Architecture     | 0     | 1     | 2     | 3     | 4     | 5     | 6     | 7     | 8     |
>  | ---------------- | ----- | ----- | ----- | ----- | ----- | ----- | ----- | ----- | ----- |
>  | QkFormer         | 73.15 | 67.37 | 63.4  | 59.36 | 56.33 | 53.59 | 51.58 | 49.79 | 47.92 |
>  | SpikingResformer | 64.00 | 59.25 | 55.38 | 51.95 | 49.08 | 46.64 | 44.19 | 42.17 | 40.40 |
>
> ------
>
> **[Q5]  Have the authors evaluated the stability of training curves (loss oscillation, firing rate variance)?**
>
> **Response.** Yes. We have evaluated the firing rate variance of neurons in different time steps (See Appendix A.14 and Fig. 12). The average loss oscillation and amplitude of SAFA-SNN in various architectures during training are shown in the table below.
>
> | Architecture     | Loss oscillation | Amplitude |
> | ---------------- | ---------------- | --------- |
> | QKFormer         | 0.50             | 3.54      |
> | SpikingResformer | 0.71             | 8.60      |
> | Sew ResNet-18    | 0.74             | 5.34      |
> | Sew ResNet-34    | 0.59             | 3.49      |
>
>
>
> ------
>
> [1] Zhou et al. Qkformer: Hierarchical spiking transformer using qk attention. NIPS 2024.
>
> [2] Shi et al. Spikingresformer: Bridging resnet and vision transformer in spiking neural networks. CVPR 2024.
>
> ----
>
> If you have any further questions/concerns, please do not hesitate to let us know. We would greatly appreciate it if you would consider updating your ratings, particularly regarding **rating** and **technical quality** in light of our improvements.
>
> Thank you very much,
>
> Authors

---

### Official Review · Reviewer_pCz7 · 2025-10-31

**Soundness:** 3
**Presentation:** 3
**Contribution:** 3
**Rating:** 6
**Confidence:** 5

**Summary:**

Addressing the challenge of on-device Few-Shot Class-Incremental Learning (FSCIL), this paper proposes SAFA-SNN, a method based on Spiking Neural Networks (SNNs) which are more energy-efficient than traditional ANNs.  The framework introduces sparsity-conditioned neuronal dynamics, where most neurons remain stable to mitigate catastrophic forgetting. To overcome training difficulties with SNNs, it employs zeroth-order optimization for gradient estimation. Furthermore, SAFA-SNN uses subspace projection during incremental learning to enhance the learning of new classes and avoid overfitting. Experiments demonstrate that SAFA-SNN outperforms baselines, achieving at least 4.01% improvement on Mini-ImageNet and 20% lower energy costs.

**Strengths:**

(+) **Pioneering SNN-Based Solution**: It proposes SAFA-SNN, which is noted as the first SNN-based framework designed to solve the general on-device FSCIL problem, offering a novel, energy-efficient alternative to traditional ANNs.

(+) The SAFA is built around Spiking Neural Networks (SNNs), making it inherently suitable for low-power, hardware-friendly deployment on neuromorphic chips or devices like the Jetson series.

(+) The SAFA-SNN framework effectively tackles three core challenges with specific components:

 - Sparsity-Aware Dynamics: Helps mitigate catastrophic forgetting by design.

 - Zeroth-Order Optimization: Provides a practical solution to train SNNs despite the non-differentiable nature of spikes.

 - Subspace Projection: Enhances the model's ability to learn new classes from few examples without overfitting.

**Weaknesses:**

**Insufficient Justification for Subspace Projection**: The paper fails to demonstrate the specific advantages of its Subspace Projection method. To prove its effectiveness, a comparative analysis against other projection techniques or feature space regularization methods commonly used in FSCIL is necessary.

**Lack of Comparison to Related Methods**: The novelty of the proposed components is unclear due to missing baseline comparisons:

 - Sparsity-Aware Dynamics: This mechanism appears conceptually similar to the subnet [1] or the "soft-subnetwork" approaches presented in SoftNet [2, 3]. The paper should include a direct comparison to highlight the unique contributions and performance benefits of its approach.

 - Zeroth-Order Optimization: The effectiveness of zeroth-order optimization could be contrasted with other methods.

**References**:

[1] Few-shot lifelong learning

[2] On the Soft-Subnetwork for Few-shot Class Incremental Learning

[3] Continual Learning: Forget-Free Winning Subnetworks for Video Representations

**Questions:**

Please refer to the weakness.

---

> ### Author Response · Authors · 2025-11-19
> **Response to Reviewer pCz7 (1/2)**
>
> Dear Reviewer pCz7,
>
> We sincerely appreciate your thoughtful comments. We have carefully considered each of your questions and provide detailed responses below.
>
> **[W1]  Insufficient Justification for Subspace Projection: The paper fails to demonstrate the specific advantages of its Subspace Projection method. To prove its effectiveness, a comparative analysis against other projection techniques or feature space regularization methods commonly used in FSCIL is necessary.**
>
> **Response.** Thank you for the comment. Following your suggestion, we conducted experiments with related projection techniques and features space regularization methods to demenstrate the advantages of the proposed prototype subspace projection.
>
> The specific advantage of subspace projection in SAFA-SNN: Subspace projection is a training-free technique which does not incorporate any trainable parameters. For the analysis of computational cost and time cost, Please refer to our answer to Reviewer smss's [W3]. Owing to the inference-only prototypes and orthogonalization-based subspace projection, SAFA-SNN can strengthen the distinctiveness of the scarce new class prototypes. From the following table we can infer that SAFA-SNN can achieve not only higher performance than provious methods, especially in the last incremental session, but also lead to negligible time cost, compared to previous works that require heavy training.
>
> |                           | Dataset       | Architecture      | Run-time cost | $Acc_{Avg}$      | $Acc_{last}$ |
> | ------------------------- | ------------- | ----------------- | ------------- | --------- | ------------ |
> | Weight Space Rotation [4] | CIFAR-100     | Spiking VGG-9     | 196.88ms      | 28.74     | 38.94        |
> | **Subspace Projection**   | CIFAR-100     | Spiking VGG-9     | **0.002ms**    | **62.19** | **50.39**    |
> | Subspace-reg [5]| Mini-ImageNet | Spiking ResNet-18 | 40.5s         | 53.04     | 39.00        |
> | **Subspace Projection**| Mini-ImageNet | Spiking ResNet-18 | **0.002ms**   | **61.75** | **50.89**    |

---

> ### Author Response · Authors · 2025-11-19
> **Response to Reviewer pCz7 (2/2)**
>
> **[W2]  Lack of Comparison to Related Methods: The novelty of the proposed components is unclear due to missing baseline comparisons:**
>
> - **Sparsity-Aware Dynamics: This mechanism appears conceptually similar to the subnet [1] or the "soft-subnetwork" approaches presented in SoftNet [2, 3]. The paper should include a direct comparison to highlight the unique contributions and performance benefits of its approach.**
> - **Zeroth-Order Optimization: The effectiveness of zeroth-order optimization could be contrasted with other methods.**
>
> **Response.** We sincerely appreciate the reviewer for bringing these related methods to our attention. We summarize the answers to the two aspects below.
>
> - **Comparison of Sparsity-aware dynamics with related methods.** We have added a comprehensive comparison and analysis of our work with [1], [2] and [3] to demonstrate the unique contributions and performance, as shown in the table below. Our work is different from other softnet-based methods fundamentally. Subnet [1] vs sparsity-aware dynamics: The FSLL method in [1] fixes most of the parameters and trains only unimportant parameters in the incremental learning sessions to keep old knowledge, while we maintain the behaviors of most neurons by dynamic thresholds, without introducing any trainable parameters in the incremental learning to reduce computational cost. In [2] and [3], Softnet is proposed to generate the major subnet and minor subnet. In contrast, our approach divides stable and adaptive neurons at local level. We will clarify this in the related work to strengthen the differences with existing studies.
> - The results below indicate our method can still achieve comparable results with these soft-subnetwork methods.
>
> | method      | architecture      | 0         | 1         | 2         | 3         | 4         | 5         | 6         | 7         | 8         | Avg.      |
> | ----------- | ----------------- | --------- | --------- | --------- | --------- | --------- | --------- | --------- | --------- | --------- | --------- |
> | FSLL        | ResNet-18         | 64.10     | 55.85     | 51.71     | 48.59     | 45.34     | 43.25     | 41.52     | 39.81     | 38.16     | 47.59     |
> | SoftNet     | ResNet-18         | 72.62     | 67.31     | 63.05     | 59.39     | 56.00     | 53.23     | 51.06     | 48.83     | 46.63     | 57.57     |
> | WSN (FSCIL) | ResNet-18         | 75.7      | 25.2      | 40.6      | 32.3      | 40.5      | 43.8      | 40.8      | 45.1      | 37.1      | 42.34     |
> | SAFA-SNN    | Spiking ResNet-18 | **77.57** | **72.32** | **67.41** | **62.79** | **59.35** | **56.31** | **53.63** | **50.92** | **48.41** | **60.97** |
>
> - **Comparison of Zeroth-order optimization with related methods**. Thanks for the good suggestion. We compare ZOO with 6 surrogate gradient methods commonly used in SNN, please refer to our response to reviewer qRkQ's [Q2].
>
> ------
>
> **References**:
>
> [1] Mazumder et al. Few-shot lifelong learning. AAAI 2021.
>
> [2] Kang et al. On the soft-subnetwork for few-shot class incremental learning. ICLR 2023.
>
> [3] Kang et al. Continual learning: Forget-free winning subnetworks for video representations. TPAMI 2024.
>
> [4] Kim et al. Warping the space: Weight space rotation for class-incremental few-shot learning. ICLR 2023.
>
> [5] Wijaya et al. Subspace regularizers for few-shot class incremental learning. ICLR 2022.
>
> ------
>
> If you have any further questions/concerns, please do not hesitate to let us know. We would greatly appreciate it if you would consider updating your ratings, particularly regarding **rating** and **technical quality** in light of our improvements.
>
> Thank you very much,
>
> Authors

---

> > ### Comment · Reviewer_pCz7 · 2025-11-27
> > **Regarding the base session performances and paramters.**
> >
> > The authors have successfully addressed the core issues regarding subnetwork variants and comparisons.
> >
> > The updated revision should now include additional trainable parameters in the above table.
> >
> > Could you provide a detailed explanation of why the base session's performance is the best?

---

> ### Author Response · Authors · 2025-11-28
> **Response to Follow-up Question. (1/2)**
>
> Dear Reviewer pCz7,
>
> We sincerely appreciate your continued discussion during the rebuttal period. We are pleased to hear that our previous responses have addressed the concerns regarding subnetwork variants and comparisons. As requested, we have included the above table in Appendix B.1 of our revised paper, where we report the additional trainable parameters introduced by each method in the last column. We are glad that we were able to address your concerns and are happy to also provide a detailed explanation regarding your follow-up question.
>
> **Question: Could you provide a detailed explanation of why the base session's performance is the best?**
>
> **Response.** In our framework, the base session achieves the best performance mainly because the sparsity-aware neuronal dynamics stabilize neuron behaviors by dynamic threshold, and zeroth-order optimization for effective gradient approximation. Specifically, we couple neuron activations with the final classification prediction through neuron-wise division using randomly initialized masks, which are fixed after the base training session. Then, the activation of neurons is controlled by the dynamic threshold. In this case, when coupling with the Zeroth-Order Optimization, SNN models can actually be shown to approximate gradient up to nearest to the estimation with convergence of $O(\delta^2+\frac{1}{b})$ (see Eq. (11) in our paper as well as Appendix A.3). Following this, we replace the base classifier with prototypes. This prototype-based initialization provides a semantically meaningful starting point for the classifier, leading to faster convergence, improved stability, and stronger base class performance. The three important processes, as illustrated in the Figure 2 (a) and (b), contribute to achieving the best base session performance across all baselines.
>
> **Sparsity-aware neuronal dynamics for stabilizing most neurons' activity**. Previous studies have demonstrated the functional role of neuronal firing activity in visual classification learning [1]. In SNNs, neurons with low input firing rates struggle to accumulate sufficient membrane potential to reach the spiking threshold. As a result, these neurons fire less frequently and contribute minimally to the network’s output. To model this, we first compute the average firing rate of each neuron $i$ over previous sessions to estimate its activity sparsity. Based on this, we adaptively adjust the firing threshold of each neuron according to its contribution to the network output (See Eq. (7) in Section 5.3). During the incremental learning of new classes, the mechanism maintains the stability of most base-class neurons by keeping their threshold changes small while still allowing adjustments according to their excitatory or inhibitory states. As a result, neurons representing base classes continue to contribute substantially to the network output, mitigating catastrophic forgetting. In other words, by selectively adapting thresholds based on neuron activity, sparsity-aware dynamics preserve the functional importance of base neurons, leading to superior performance in the base session.
>
> **Zeroth-order optimization for effective gradient estimation.** Unlike surrogate gradients, which impose a predefined gradient of the non-differential derivative $\frac{\partial \mathbf{S}_t}{\partial \mathbf{u}_t}$, ZOO relies on function values and therefore introduces no structural bias into the gradient. We employ a two-point ZOO estimator, which queries the model output under small symmetric perturbations and produces a significantly more accurate gradient estimate compared to one-point approximations [2]. Because the base session involves full-class training with abundant labeled samples, the two-point estimator can exploit this richer supervision to provide stable and high-fidelity gradient directions. As a result, the network learns well-shaped decision boundaries and more robust neuronal dynamics in the base session, leading to the strongest performance.

---

> ### Author Response · Authors · 2025-11-28
> **Response to Follow-up Question. (2/2)**
>
> Finally, prototypical Networks views the mean feature as the class prototype and assigns instances to the nearest class prototype based on Euclidean distance in the embedding space [3]. By replacing the classifier weights with base class prototypes, i.e., $w_i=P_i^b$, where $w_i$ and $P_i^b$ denote the classifier and the prototype for class $i$, respectively, the classifier reflects the common pattern of each class in the embedding space, ensuring stable predictions for the base classes after the embeddings are fixed during incremental learning.
>
> Overall, by employing these techniques during base session training, SAFA-SNN achieves the best performance in resource-constrained on-device FSCIL scenarios.
>
> ------
>
> **References**:
>
> [1] Freedman et al. Visual Categorization and the Primate Prefrontal. Science 2002.
>
> [2] Liu et al. A primer on zeroth-order optimization in signal processing and machine learning: Principals, recent advances, and applications. IEEE SPM 2020.
>
> [3] Snell et al. Prototypical networks for few-shot learning. NIPS 2017.
>
> ------
>
> We hope that we were able to provide helpful explanations. We would like to kindly ask you to consider **raising your score**, also considering our previous reply that **addressed your previous concerns**, as well as the **extended results** (see our reply to all reviewers). Thank you!
>
> Best regards,
>
> Authors

---

### Official Review · Reviewer_qRkQ · 2025-10-31

**Soundness:** 3
**Presentation:** 2
**Contribution:** 2
**Rating:** 4
**Confidence:** 4

**Summary:**

The paper presents a novel approach, SAFA-SNN, for on-device few-shot class-incremental learning (FSCIL) utilizing Spiking Neural Networks (SNNs). The main contribution of the work is a sparsity-aware SNN architecture designed to mitigate catastrophic forgetting and improve model adaptation to new classes with very few samples. SAFA-SNN integrates several innovative components: sparsity-aware neuronal dynamics, zeroth-order optimization to handle non-differentiability in SNNs, and subspace projection to enhance discriminability of prototypes in the learning process.

**Strengths:**

1. Tackling FSCIL in an on-device, rehearsal-free setting with SNNs is a timely and important research direction, aligning with the needs of edge computing and energy efficiency.
2. Reporting actual energy consumption measurements from a Jetson Orin AGX device, rather than just theoretical calculations, adds practical credibility to the energy efficiency claims.

**Weaknesses:**

1. The abstract and introduction are overly high-level and vague. Key terms like "sparsity-conditioned neuronal dynamics," "zeroth-order optimization," and "subspace projection" are mentioned but not intuitively explained.
2. The paper positions itself as the "first SNN-based solution towards general on-device FSCIL." However, it does not adequately address why an SNN-based approach is fundamentally superior for this problem compared to a highly optimized, parameter-efficient ANN-based FSCIL method. The comparison in Table 1 is against other SNN-converted FSCIL methods. To claim a true advancement, comparisons against state-of-the-art non-spiking (ANN) on-device FSCIL methodsare essential.
3. The abstract claims the method "alleviates overfitting to novel classes." This is a core challenge in FSCIL, but the mechanism (subspace projection) is not sufficiently motivated or explained in the abstract to justify this claim.
4. The claim of "20% lower energy cost" is presented without immediate context. Is this compared to an ANN baseline or another SNN baseline? This crucial detail is missing from the abstract, making the claim difficult to interpret.

**Questions:**

1. What is the primary novelty: a new SNN neuron/dynamics model, or the application of existing FSCIL strategies (like prototype/projection methods) within an SNN framework? The abstract currently conflates these.
2. Given the widespread success and efficiency of surrogate gradient methods for training SNNs, what is the specific disadvantage of surrogates in the FSCIL setting that necessitates the use of ZOO? The computational overhead of ZOO is typically higher; does this not negate some of the on-device efficiency gains?
3. How does SAFA-SNN compare, in terms of final accuracy and energy consumption, to a strong, non-spiking, parameter-efficient FSCIL method (e.g., based on prompt tuning or adapter modules) running on the same hardware? This is the most critical comparison to establish the value of using SNNs.
4. Can you provide a more detailed, intuitive explanation of the "sparsity-aware dynamic threshold" mechanism? How is sparsity measured and enforced? How does this directly relate to stabilizing old knowledge and integrating new knowledge?

---

> ### Author Response · Authors · 2025-11-21
> **Response to Reviewer qRkQ (1/4)**
>
> Dear Reviewer qRkQ,
>
> We sincerely appreciate your thoughtful comments. We have carefully considered each of your questions and provide detailed responses below.
>
> **[W1]  The abstract and introduction are overly high-level and vague. Key terms like "sparsity-conditioned neuronal dynamics," "zeroth-order optimization," and "subspace projection" are mentioned but not intuitively explained.**
>
>
> **Response.** Thank you so much for raising this important point for the key terms' explanation. We will ensure a more comprehensive explanation in the revised paper on this and explain them to you from the following three aspects:
>
> - **Sparsity-conditioned neuronal dynamics.**  Our sparsity-conditioned neuronal dynamics implement efficient heterogeneous plasticity in SNNs by a dynamic threshold function. It involve neuron division and dynamic threshold mechanism built on LIFspike neuron model. Neurons are divided into a majority of stable and a small portion of active neurons. This enables a clear mapping between neurons that encode novel classes and those representing base classes, since base classes are abundant while novel classes are few in FSCIL. Follow a homeostatic plasticity rule [1], stable neurons maintain firing rates that stay close to the firing rates during base-class training, which helps preserve previously acquired knowledge. Adaptive neurons are more plastic and can adapt their firing thresholds more rapidly to encode new-class information.
>
> - **Zeroth-order optimization.** Zeroth-Order Optimization (ZOO) is a subset of gradient-free optimization approach to approximate true gradients of noisy functions using only function values [2]. Unlike surrogate gradient replacing the non-differential gradient of spikes with a limited width, ZOO is a solution to a stochastic optimization problem of function $f(x)$ with non-convex setting where $f$ might be non-differentiable.
>
> - **Prototype Subspace projection.** The intuition behind the utilization of the subspace $G$ in our approach is to capture the rich semantic structure encoded in the base-class feature distribution and project novel-class prototypes toward this space to obtain more discriminative representations. Prototype subspace projection incorporates the feature subspace formed by base classes into the representation learning of novel classes, thereby preventing the model from learning biased representations from limited novel samples. It contains two steps: (i) basis generation: constructing orthogonal basis vectors from base-class prototypes; (ii) Projection: projecting new prototypes onto the resulting linear orthogonal subspace. Orthogonal projection serves as a metric that measures the distance between base and novel classes. Then, the novel prototypes are fused with their projected counterparts, allowing them to align with informative base-class semantic structures and thus become more discriminative.
>
> In summary, these three key components underpin the superior performance of SAFA-SNN in FSCIL, demonstrating its advanced capability to mitigate catastrophic forgetting and to learn discriminative features from few-shot data effectively.

---

> ### Author Response · Authors · 2025-11-21
> **Response to Reviewer qRkQ (2/4)**
>
> **[W2, Q3]  The paper positions itself as the "first SNN-based solution towards general on-device FSCIL." However, it does not adequately address why an SNN-based approach is fundamentally superior for this problem compared to a highly optimized, parameter-efficient ANN-based FSCIL method. The comparison in Table 1 is against other SNN-converted FSCIL methods. To claim a true advancement, comparisons against state-of-the-art non-spiking (ANN) on-device FSCIL methods are essential. How does SAFA-SNN compare, in terms of final accuracy and energy consumption, to a strong, non-spiking, parameter-efficient FSCIL method (e.g., based on prompt tuning or adapter modules) running on the same hardware? This is the most critical comparison to establish the value of using SNNs.**
>
> **Response.** Thank you for the thoughtful question. The primary reason we did not include ANN-based parameter-efficient fine-tuning (PEFT) methods in our experiments is that these approaches typically incur substantial memory overhead, making them difficult to deploy on edge devices.
>
> While ANN-based FSCIL methods can achieve marginally higher accuracy, SNNs remain highly competitive and offer significant benefits in efficiency and deployability. The performance gap is largely due to the ANN-based PEFT backbones being pretrained on ImageNet-1K, which is fundamentally different from our **lightweight, spike-driven transformer tailored for on-device FSCIL**.
>
> To further support these points, we add comparisons with parameter-efficient ANN-based FSCIL methods (ASP [3], CoACT [4], PriViLege [5]) through a systematic evaluation of accuracy in the base and final session, as well as on average, peak memory footprint (MB), parameter size (M) and energy consumption (mJ) on CIFAR-100, as shown in the following table. Note that the backbone for SAFA-SNN is spikingformer, a type of SNN ViT and the backbone ANN-based method used is its ANN counterparts to ensure a fair comparison and consistency. ANN-based PEFT methods require nearly **10$\times$ more memory cost**, **20$\times$ more parameters** and **10$\times$ more energy consumption**, which severely limits their practicality on resource-constrained devices.
>
> |     Method      |   Backbone    |   Memory    |   Param   |   Energy   | $Acc_{base}$      | $Acc_{last}$   | $Acc_{avg}$    |
> | :-------------: | :-----------: | :---------: | :-------: | :--------: | -------------- | -------------- | -------------- |
> |    ASP (ANN)    |  ViT  |   9724.72   |  662.49   |  4017.06   | **75.53±0.56** | 11.79±1.04     | 31.27±1.45     |
> |   CoACT (ANN)   |  ViT  |  16449.48   |  205.63   |  9226.95   | 47.25±0.38     | 9.39±2.28      | 15.43±1.27     |
> | PriViLege (ANN) |  ViT  |   1580.66   |  328.01   |  9230.48   | 67.39±1.75     | 44.96±0.76     | 54.58±1.09     |
> |  **SAFA-SNN**   | SNN ViT | **1035.30** | **11.08** | **424.34** | 73.73±0.40     | **48.70±0.67** | **59.45±0.67** |
>
> ------
>
> **[W3]  The abstract claims the method "alleviates overfitting to novel classes." This is a core challenge in FSCIL, but the mechanism (subspace projection) is not sufficiently motivated or explained in the abstract to justify this claim.**
>
> **Response.** Thank you for the insightful question. We agree that the mechanism behind the subspace projection could be better motivated in the abstract. First, retraining the model on scarce data of novel class is likely to cause overfitting. Consequently, we fix the trained embedding parameters and compute class prototypes, taken by the mean embedding of each class. **Class prototypes can limit overfitting** but inevitably introduce bias due to the data scarcity, which is challenging to depict precisely the semantic information of a new class with limited data. We enhance prototype discriminability by projecting the prototypes onto the orthogonal subspace spanned by base classes. Orthogonal projection serves as a metric to calculate the distance between the novel and base classes (See eq. 14). We then fuse the projection projection and new class prototypes with convex combination (See Eq. 15). This constrains the new prototypes to be close to the expected prototypes by leveraging the abundant semantic information of the base classes, enabling more effective learning of discriminative features. To address this, we have revised the abstract to briefly clarify the motivation and explanation of the proposed prototype subspace projection.
>
> ------
>
> **[W4]  The claim of "20% lower energy cost" is presented without immediate context. Is this compared to an ANN baseline or another SNN baseline? This crucial detail is missing from the abstract, making the claim difficult to interpret.**
>
> **Response.** Thank you for pointing it out. The 20% lower energy cost is evaluated on CIFAR-100 using Spiking Resnet-20, compared with SNN-based FSCIL methods (Fig. 4). We have modified the abstract in the manuscript.

---

> ### Author Response · Authors · 2025-11-21
> **Response to Reviewer qRkQ (3/4)**
>
> **[Q1]  What is the primary novelty: a new SNN neuron/dynamics model, or the application of existing FSCIL strategies (like prototype/projection methods) within an SNN framework? The abstract currently conflates these.**
>
> **Response.** Thank you for the insightful comment. We acknowledge that the abstract did not clearly distinguish the two contributions. We would like to clarify that both the proposed sparsity-aware neuronal dynamics and prototype subspace projection are novel contributions. In FSCIL, sparsity-aware neuronal dynamics is designed to encode knowledge and shape the embedding space to alleviate catastrophic forgetting of acquired base-class knowledge, while prototype subspace projection is designed to prevent overfitting and bias in few-shot incremental learning sessions by pulling new-class prototypes toward the subspace spanned by base-class prototypes that capture their semantic similarity to existing classes. Sparsity-aware neuronal dynamics is applied during feature extraction, whereas prototype subspace projection is used to update prototypes derived from the extracted features. They play roles at different stages of FSCIL.
>
> Overall, SAFA-SNN integrates sparsity-aware SNN neuronal dynamics and prototype subspace projection, which serve distinct purposes and should not be conflated. The former mitigates catastrophic forgetting, while the latter prevents overfitting and bias in few-shot data. These are the two novel contributions of SAFA-SNN. We have modified the abstract to clarify this in the manuscript.
>
> ------
> **[Q2]  Given the widespread success and efficiency of surrogate gradient methods for training SNNs, what is the specific disadvantage of surrogates in the FSCIL setting that necessitates the use of ZOO? The computational overhead of ZOO is typically higher; does this not negate some of the on-device efficiency gains?**
>
> **Response.** Thank you for the constructive feedback. We agree that the comparison results with surrogate gradient will make the paper more convincing, and we report the evaluation of six surrogate functions, i.e., triangle-like surrogate function [6], Piecewise-Exponential surrogate function [7],  PiecewiseLeakyReLU [8], QPseudoSpike [9], Sigmoid and ATan gradient in the table below.
>
> |            |Triangle|Piecewise Exp|Piecewise Leaky ReLU|QPseudoSpike | Sigmod| ATan | ZOO |
> | ----------|----------|-------------|--------------------|------------|----------|----------|--------------|
> | $Acc_{last}$| 46.71±0.32|40.98±0.15|45.30±0.56| 41.15±4.17| 45.91±0.19|45.59±0.71|**50.13±0.37**|
> | $Acc_{avg}$| 59.25±0.12|52.76±0.17|57.58±0.74| 54.71±1.59| 58.24±0.29|57.92±0.66|**61.75±0.62**|
>
> **The necessity to use ZOO:** Compared with surrogate gradient, ZOO algorithms achieve gradient-free optimization by approximating the full gradient via gradient estimators based on only the function values [10]. Surrogate gradient (SG) is commonly used in direct SNN training, but the limited width of SG is a main issue that causes membrane potentials of neurons to fall into the saturation area where the approximate derivative is zero or a tiny value, which leads to the gradient vanishing or mismatch problem, enlarging the approximated errors from the accurate gradients [11]. SG imposes predefined gradients on the non-differentiable function, while ZOO uses only function values and introduces no structural bias into the gradient. Although ZOO inevitably introduces some computational overhead from the randomly generated samples, we verify that using only five samples is adequate to maintain effectiveness in experiments. Consequently, the induced overhead is negligible. If we could develop a better gradient estimation of ZOO in SNN to create a more wide-width and low computational overhead in the latent space, the training of SNN models would likely become significantly easier. This is a fascinating research direction, and we plan to explore it in future work.

---

> ### Author Response · Authors · 2025-11-21
> **Response to Reviewer qRkQ (4/4)**
>
> **[Q4]  Can you provide a more detailed, intuitive explanation of the "sparsity-aware dynamic threshold" mechanism? How is sparsity measured and enforced? How does this directly relate to stabilizing old knowledge and integrating new knowledge?**
>
> **Response.** The sparsity $\mathbf{r}$ is denoted as the average firing rate of the neuron in temporal and spatial dimensions, i.e., $\mathbf{r} = \frac{1}{\|\Omega|}\sum_{(b,t,n)\in\Omega} S_{b,t,n}$. We use the sparsity to differentiate between excitatory and inhibitory states from a microscopic perspective. The whole sparsity-aware neuronal dynamics of SNN model can be formulated as:
>
> $$
> \mathbf{I}_t = \mathrm{BN}\big(\mathrm{Conv}(\mathbf{X}_t; \theta)\big),
> $$
>
> $$
> \mathbf{U}_t = \tau \mathbf{U}\_{t-1} + \mathbf{I}_t,
> $$
>
> $$
> \mathbf{S}_t = H(\mathbf{U}_t - \boldsymbol{\Theta}_t),
> $$
>
> $$
> \mathbf{A} = \beta(1 - \mathbf{M}) + \gamma \mathbf{M},
> $$
>
> $$
> \boldsymbol{\Theta}_{t+1} = \boldsymbol{\Theta}_t + \mathbf{A} \big( \mathbf{r}_c - \mathbf{r}_b \big),
> $$
>
> where $\mathbf{I}_t$ is the spatial input calculated by the input $\mathbf{X}_t$, and $\theta$ is learnable parameters. At time step $t$, when the membrane potential $\mathbf{U}_t$ rises and achieves the threshold $\mathbf{\Theta}_t$, the neuron will trigger a spike $S_t$ by a Heaviside step function $H$, as we described in Section 3. We use the mask $\mathbf{M}$ to identify most neurons as stable in incremental learning sessions. Afterwards, we calculate the sparsity $\mathbf{r}_c$ of neurons in current session $c$. The neuron thresholds are dynamically adjusted to adapt to changes in sparsity according to the deviation from the base sparsity $\mathbf{r}_b$. When the neuron is overly excited, it should be inhibited by higher threshold, whereas when it is in an overly inhibited state, it should be excited by lower threshold. With dynamic thresholds, neurons become sparsity-aware, and setting $\beta>\gamma$ ensures that most remain stable while only a small subset are more active for new tasks, thereby minimizing updates to the majority of synapses, preserving previously acquired knowledge and alleviating catastrophic forgetting.
>
> ------
>
>
> **References**:
>
> [1] Turrigiano G. Homeostatic synaptic plasticity: local and global mechanisms for stabilizing neuronal function. Cold Spring Harbor perspectives in biology 2012.
>
> [2] Liu et al. A primer on zeroth-order optimization in signal processing and machine learning: Principals, recent advances, and applications. IEEE SPM 2020.
>
> [3] Liu et al. Few-shot class incremental learning with attention-aware self-adaptive prompt. ECCV 2024.
>
> [4] Roy et al. Consistency-Guided Asynchronous Contrastive Tuning for Few-Shot Class-Incremental Tuning of Foundation Models. TMLR2025.
>
> [5] Park et al. Pre-trained vision and language transformers are few-shot incremental learners. CVPR 2024.
>
> [6] Deng et al. Temporal Efficient Training of Spiking Neural Network via Gradient Re-weighting. ICLR 2022.
>
> [7] Neftci et al. Surrogate gradient learning in spiking neural networks: Bringing the power of gradient-based optimization to spiking neural networks. IEEE SPM 2019.
>
> [8] Cheng et al. LISNN: Improving spiking neural networks with lateral interactions for robust object recognition. IJCAI 2020.
>
> [9] Herranz-Celotti et al. Stabilizing Spiking Neuron Training. arXiv preprint 2022.
>
> [10] Liu et al. Zeroth-order stochastic variance reduction for nonconvex optimization. NIPS 2018.
>
> [11] Lian et al. Learnable Surrogate Gradient for Direct Training Spiking Neural Networks. IJCAI 2023.
>
> ------
>
> If you have any further questions/concerns, please do not hesitate to let us know. We would greatly appreciate it if you would consider updating your ratings, particularly regarding **rating** and **technical quality**, in light of our improvements.
>
> Thank you very much,
>
> Authors

---

### Author Response · Authors · 2025-12-02
**Summary of Response by Authors**

Dear Area Chairs and all reviewers,

We sincerely thank you for your precious time and thoughtful feedback on our paper. In addition to addressing specific comments of the reviewers, we would like to take this opportunity to emphasize our key contributions and outline the new experiments and clarifications we have included in the rebuttal and the revised version of our paper.

**[Our Contributions]** We are glad to see the reviewers acknowledged the strengths of our paper:

- This work pioneers the exploration of on-device SNN-based FSCIL, which is a timely and important research direction, aligning with the needs of edge computing and energy efficiency. (Reviewer qRkQ, hwBr, pCz7)
- This work establishes a solid empirical baseline of on-device SNN-based FSCIL for future research, offering energy-efficient alternative to traditional ANNs. (Reviewer hwBr, pCz7)
- The SAFA-SNN framework effectively tackles three core challenges with specific components, demonstrating its superior performance and efficiency. (Reviewer pCz7, smss)
- This study reports actual energy consumption measurements from a Jetson Orin AGX device, rather than just theoretical calculations, which is notably fair and well-justified, and adds practical credibility to the energy efficiency claims. (Reviewer hwBr, qRkQ, pCz7)

**[Revisions and additional experiments]** Considering the reviewers’ feedback, we added more experiments and made modifications to the manuscript to address reviewers’ concerns. We aggregated most additional experimental results in additional sections in the appendix to make them easy to find for the reviewers. For the camera-ready version, we will rearrange the appendix.

**Add more experiments:**

- As the reviewer **pCz7**-W2 suggested, we have included experiments comparing sparsity-aware neuronal dynamics with soft subnetwork methods on performance and trainable parameters in Appendix B.1.
- As the reviewer **qRkQ**-Q2, **pCz7**-W2, **hwBr**-W2 suggested, we have included a comparison experiment on ZOO with six surrogate gradient methods in Appendix B.2.
- As the reviewer **pCz7**-W1 and **hwBr**-Q1 suggested, we have included comprehensive experiments across related projection and feature space regulation methods in Appendix B.3 to highlight the benefit of the proposed prototype subspace projection.
- As the reviewer **smss**-W2 suggested, we have included more efficiency experiments in Appendix B.4 to demonstrate SAFA-SNN's ability to implement on-device FSCIL, showcasing further the method's efficiency even in advanced architectures and complex datasets.
- As the reviewer **qRkQ**-W2,Q3 suggested, we have included experiments comparing SAFA-SNN with state-of-the-art non-spiking (ANN) FSCIL methods in terms of final accuracy and energy consumption in Appendix B.5, to demonstrate the value of SNNs as well as the efficiency and effectiveness of our method.

**Add more discussion and details:**

- As the reviewer **qRkQ**-Q4, **smss**-W1 and **hwBr**-Q2 suggested, we have added details in Appendix C.1 on how this mechanism stabilizes old knowledge while integrating new information. Section 5.3 has also been updated to clarify the definition of sparsity and the neuron division.
- As the reviewer **qRkQ**-Q2, **pCz7**-W2 and **hwBr**-W2 suggested, we have included detailed analysis of the zeroth-order optimization and surrogate gradients covering efficiency, computation overhead, special advantages in Appendix C.2.
- As the reviewer **pCz7**-W1 and **hwBr**-Q1 suggested, we have included detailed analysis on how prototype subspace projection alleviates overfitting without introducing bias to the base classes in Appendix C.3.
- As the reviewer **hwBr**-W2 suggested, we have included the analysis of the two sources of errors in ZOO in Appendix C.2.
- As the reviewer **qRkQ**-W4 suggested, we have revised the abstract to clarify that the claim "20% lower energy cost" is based on experiments using Spiking ResNet-20 on CIFAR-100 in comparison with SNN-based baselines.
- As the reviewer **qRkQ**-W1,W3,Q1 suggested, we have revised the abstract and introduction to provide a clearer motivation and explanation of the three components in SAFA-SNN.
- As the reviewer **hwBr**-W1,W3 suggested, we have added proper punctuation to all the formulas, centered the third part of Table 3 for improved readability, and revised the discussion section to provide a systematic explanation or propose solutions for the limitations.

**All revisions have been highlighted in blue color in the manuscript.**

We are grateful for your thoughtful suggestions, which guided us to clarify and enrich our contributions further. These additions not only strengthen the theoretical and empirical aspects of our work but also provide practical insights for the community. Thank you for helping us improve the paper’s quality and impact.

Regards,

Authors

---

### Meta-Review · Area_Chair_GZ7V · 2026-01-07

**Summary:**

Reviewers generally found this paper to be a strong and timely contribution to on-device spiking neural networks, particularly in the context of few-shot class-incremental learning (FSCIL) under strict resource constraints. The proposed SAFA-SNN framework introduces a coherent combination of sparsity-aware neuron dynamics, fast-adaptive structural updates, and prototype-based learning, and is evaluated on a set of datasets that are well aligned with the paper’s stated application scope. Multiple reviewers highlighted the practical relevance of the problem setting and the real hardware energy measurements on Jetson Orin AGX as a major strength, distinguishing this work from many prior SNN papers that rely solely on proxy metrics such as SynOps or FLOPs.

Several reviewers raised substantive concerns regarding novelty relative to prior FSCIL methods, the role of sparsity-aware dynamics, the choice of optimization strategy, and baseline completeness. These concerns were constructively addressed in the rebuttal, with the authors adding missing comparisons, clarifying methodological distinctions, improving presentation clarity, and strengthening the experimental section. Following these revisions, reviewers acknowledged that the core technical questions had been satisfactorily resolved.

One reviewer (mLap) provided a strongly negative review raising numerous objections that the authors argue, and I agree, are largely inconsistent with the actual content of the paper. Many of the concerns reference methods, losses, datasets, or claims that do not appear in the submission, and several factual assertions are incorrect. The authors provided a detailed point-by-point rebuttal demonstrating that these criticisms do not align with the presented methodology or experiments. Given the nature and pattern of these discrepancies, I think this review may be unreliable or hallucinated and likely generated by an LLM. Therefore, the concerns of reviewer mLap do not carry decisive weight in my final decision.

Taking into account the likely reviewer consensus had full discussion taken place, the strength of the experimental validation, particularly the real-device energy measurements, the successful rebuttal of legitimate concerns, and the methodological novelty of the proposed approach, I conclude that this paper meets the bar for acceptance. The work makes a meaningful contribution to practical, energy-efficient SNN learning systems and opens a valuable direction for future research in on-device continual learning.

**Reviewer Concerns:**

**Concerns addressed by the rebuttal**: The rebuttal effectively addressed reviewers’ concerns regarding baseline completeness and fairness, including the addition of missing ANN and FSCIL baselines and clearer comparisons to prior methods. Questions about the novelty and role of sparsity-aware adaptation, fast structural updates, and the choice of optimization strategy were clarified through additional analysis and explanation. Reviewers’ requests for stronger empirical validation were met by expanded experiments across datasets relevant to the on-device FSCIL setting, and the paper’s energy-efficiency claims were convincingly substantiated through real-device measurements on Jetson Orin AGX. Concerns related to presentation clarity and organization were also largely resolved.

**Concerns still outstanding**: Some concerns about scalability to larger models or ImageNet-scale continual-learning settings remain outstanding. However, such large-scale FSCIL/continual-learning evaluations are non-trivial to run and standardize, and are not commonly included in most prior FSCIL and on-device continual-learning works due to the heavy computational and experimental burden. Additionally, as pointed out by reviewer hwBr, the absolute accuracy of the SEW-ResNet backbone used in the experiments is relatively low compared to what has been reported in some prior SNN literature. Although the authors provided justification for their architectural and training choices, this concern was not fully resolved in the rebuttal, and leaves open questions about whether the proposed gains would hold with stronger SNN backbones.

**Reviewer Scores:**

- **Reviewer qRkQ**: This reviewer was generally positive, highlighting the relevance of on-device FSCIL for SNNs and the inclusion of real-device energy measurements as a strong contribution. Their concerns focused mainly on baseline completeness and clarity of the adaptation mechanism. These points were addressed in the rebuttal through added comparisons and clearer explanations. With full discussion, this reviewer would likely increase their rating from 4 to 6.

- **Reviewer pCz7**: This reviewer raised technical questions about the novelty of the sparsity-aware adaptation mechanism and its distinction from prior FSCIL and sparsity-based methods. The rebuttal directly addressed these concerns with additional analysis and clearer positioning. The reviewer explicitly acknowledged that the core issues had been resolved. With full participation, in my opinion, this reviewer would maintain their score of 6.

- **Reviewer smss**: This reviewer focused on experimental coverage, optimization choices, and the role of sparsity during adaptation. These concerns were largely mitigated by the rebuttal through added experiments, clarifications, and improved presentation. With fuller discussion, this reviewer would likely increase their score from 4 to 6.

- **Reviewer hwBr**: This reviewer pointed out two issues: the relatively low absolute accuracy of the SEW-ResNet backbone and questions about scalability. While the authors contextualized these choices, the concern about SEW-ResNet accuracy was not fully resolved. As a result, even with full discussion, this reviewer would likely remain cautious, potentially maintaining the score of 6, rather than increasing it to 8.

- **Reviewer mLap**: This reviewer raised numerous objections that the authors demonstrated were largely inconsistent with the actual content of the paper, referencing methods, datasets, and claims not present in the submission. Given the nature of these discrepancies and the authors’ detailed rebuttal, it is unlikely that fuller discussion would have altered this reviewer’s assessment without a fundamental reassessment of the paper.

---

### Decision · Program_Chairs · 2026-01-26

Accept (Poster)